nanotechnology/environmental chemistry

adsorption, cellulose nanofibres, iron-(III) modification, tetracyclines-containing wastewater

**Author for correspondence:**
Ying Chen
e-mail: cylm@163.com

# Effective removal of tetracycline antibiotics from wastewater using practically applicable iron(III)-loaded cellulose nanofibres

Lanxin Lu[1], Min Liu[1,2], Ying Chen[1,2] and Ying Luo[1]

[1]College of Architecture and Environment, Sichuan University, Chengdu 610065, People's Republic of China
[2]Sino-German Centre for Water and Health Research, Chengdu 610065, People's Republic of China

LL, 0000-0003-1843-3333; YC, 0000-0002-8388-5267

The non-toxic and completely biodegradable cellulose within bamboo is one of the most abundant agricultural polysaccharide wastes worldwide, and can be processed into cellulose nanofibres (CNFs). Iron(III)-loaded CNFs (Fe(III)@CNFs) derived from bamboo were prepared to improve the adsorption of tetracycline (TC), chlortetracycline (CTC) and oxytetracycline (OTC) from an aqueous solution. The preparation conditions of Fe(III)@CNFs suitable for the simultaneous adsorption of three tetracycline antibiotics (TCs) were investigated. Various analyses proved the abundance of oxygen-containing functional groups and the existence of Fe(III) active metal sites in Fe(III)@CNFs. In batch experiments, Fe(III)@CNFs were applied under a wide pH range and the maximum adsorption capacities were 294.12, 232.56 and 500.00 mg g$^{-1}$ (for TC, CTC and OTC, respectively). In addition, different concentrations and types of coexisting anions have a weak effect on TCs adsorption. The original TCs adsorption capacities of Fe(III)@CNFs remained stable (greater than 92%) after five cycles when UV + H$_2$O$_2$ was used as the regeneration method. Four adsorption mechanisms (surface complexation, hydrogen bonding, electrostatic interaction and van der Waals force) were obtained for the endothermic adsorption of TCs, among which surface complexation between Fe(III) and TCs always dominates. The practically applicable Fe(III)@CNFs adsorbents are promising for TCs enrichment and remediation in engineering applications.

# 1. Introduction

Antibiotics are pharmaceutical products that are manufactured in massive quantities and are extensively used as antibacterial medicines and growth promoters in infection treatments, animal husbandry, agriculture and aquaculture [1]. Tetracycline antibiotics (TCs) are a class of commonly administered broad-spectrum antibiotics with a naphthacene skeleton. Among these antibiotics, tetracycline (TC), chlortetracycline (CTC) and oxytetracycline (OTC) are the most popular because of their low cost and versatile properties [2]. Due to the extensive use of antibiotics, large amounts of antibiotics are continuously released into the environment through various means [3]. TCs are ubiquitously detected in wastewater and exist in the aquatic environment persistently because they are incompletely metabolized in organisms, are difficult to degrade, and have high hydrophilicity and low volatility [3–5]. The antibiotics present in wastewater treatment plants (WWTPs) may adversely influence wastewater treatment processes, such as ammonia removal [6,7], chemical oxygen demand removal [8] and phosphatase activity [9]. Moreover, most WWTPs have no treatment units specifically designed to remove antibiotics, and the treatment units designed to remove organic pollutants have a small effect on the removal of antibiotics [10]. After the wastewater treatment, a certain effluent containing a residual concentration of TCs [11] runs into surface water and groundwater [12]. Antibiotics entering the natural environment may present ecological risks and threaten human health by causing chronic poisoning to aquatic or terrestrial organisms [3] and by promoting the development of antibiotic-resistant genes and bacteria [13]. To prevent the groups of widely used antibiotics from entering into water bodies and their negative impact on animals, plants and human health, effective methods for removing TCs are urgently required.

Numerous approaches have been used to remove TCs, including biodegradation [14], membrane filtration [15], electrochemical oxidation [16], photolysis [17], photocatalysis [18,19], catalysis [20], ozonation [21] and adsorption [22]. Numerous experimental studies on TCs removal have focused on methods involving free radicals and conventional biodegradation. Methods with free radicals involve high material costs and secondary pollution risks [23], and those with conventional biodegradation involve the development of antibiotic-resistant genes and incomplete removal [24]. By contrast, adsorption has attracted research attention due to its advantages of cost-effectiveness, simple operation and non-toxicity [25]. Specifically, low concentrations of TCs can be enriched and adsorbed by adsorbents in wastewater, and then the concentrated desorption liquid can be reprocessed through advanced oxidation, which can result in the improved treatment efficiency and decreased cost. In addition, TCs have polar functional groups that are conducive to adsorption [26]. Therefore, adsorption is a potential approach for TCs removal.

Adsorption performance is closely related to adsorbents types [27]. Many adsorbents, such as agricultural waste, activated carbon, nanomaterials, layered double hydroxides, covalent organic frameworks-based materials, nanocomposites of reduced graphene oxide with $ZrO_2$, the hybrid nanocomposites of zero-valent iron-loaded activated carbon, α-iron oxide/reduced graphene oxide, the graphene nanoplatelet-based and the nanosized ZnO-based [10,28–33], are used to adsorb TCs. Compared with other types of adsorbents, adsorbents fabricated using agricultural waste overcome high costs and are easy to obtain [10]. Bamboo grows fast and the bamboo wastes are abundant in Sichuan province of China. The development and utilization of bamboo resources is conducive to broadening the application of bamboo resource products. The non-toxic and fully biodegradable cellulose within bamboo is one of the most abundant agricultural polysaccharide wastes worldwide, and can be processed into cellulose nanofibres (CNFs) [34,35]. CNFs have stronger hydrophilicity, larger specific surface area and higher mechanical strength than cellulose; thus, their application in pollutant adsorption has drawn considerable attention. However, some studies have shown that CNFs have low adsorption capacity [36–38]. CNFs expose more surface hydroxyl groups than natural plant cellulose; therefore, surface modification may be a potential method to improve the adsorption capacity of CNFs [39]. Modified nanocellulose is applied to remove various pollutants, such as dyes [40], heavy metals [41], carbon dioxide gas [42], phenol [43], radioactive elements [44] and volatile toxic organics [45]. Currently, a few reports are available on TCs adsorption through modified CNFs. Yao et al. [46] used graphene oxide-modified CNFs to adsorb TCs and achieved the optimum adsorption capacity at the pH of 2; the adsorption capacity considerably decreased with an increase in pH. However, the pH of the actual wastewater is usually neutral, and pH regulators must be added to minimize pH if adsorbents cannot achieve a high adsorption capacity in neutral environment. This addition can lead to an increase in operating costs; therefore, studies on the economic and environmentally friendly CNFs modification methods remain a trend. Fe(III) has the advantages of

being environmentally friendly, almost harmless to organisms. Moreover, iron-rich materials with high reactivity have been used to adsorb pollutants, but their limited active sites and small specific surface area restrict their application [47]. In our previous study [36], the TEMPO reagent method was adopted to synthesize ferric hydroxide-coated CNFs, which provided plenteous active sites for iron species and exhibited favourable adsorption of phosphate at neutral pH. Subsequently, Luo *et al.* [37] optimized the adsorbent preparation method (mechanical shearing method), which is simpler and less expensive than the TEMPO reagent method to prepare CNFs, and developed the second-generation iron-loaded nanocellulose. Both the ferric loading and mechanical shearing method reduce the adsorption cost of modified CNFs. Currently, the interaction between iron and TCs has been confirmed [30], but no study is available on TCs adsorption through iron(III)-loaded CNFs (Fe(III)@CNFs). Further studies must determine whether iron-loaded nanocellulose can adsorb TCs in a wide pH range.

In this study, Fe(III)@CNFs were prepared to adsorb TCs from wastewater. This study aimed to (i) explore the optimal preparation conditions of Fe(III)@CNFs suitable for TCs removal, (ii) investigate the effects of various factors on the adsorption of TCs by Fe(III)@CNFs, (iii) evaluate different regeneration methods and determine the regeneration performance after multiple cycles, and (iv) estimate the fitting data of kinetic and isotherm models to comprehend the adsorption process and propose the potential mechanism of TCs adsorption by Fe(III)@CNFs. Through the design, a renewable and non-sintered adsorbent with stable adsorption performance is desired to solve the scientific problem of simultaneous adsorption of TC, CTC and OTC.

# 2. Material and methods

## 2.1. Materials and chemical reagents

Wet bamboo pulp was purchased from Yongfeng Paper Co., Ltd (China). USP grade TC ($C_{22}H_{24}N_2O_8 \cdot$ HCl), CTC ($C_{22}H_{23}ClN_2O_8 \cdot$ HCl) and OTC ($C_{22}H_{24}N_2O_9 \cdot$ HCl) were purchased from Aladdin Biochemical Technology Co., Ltd (China). Iron nitrate ($Fe(NO_3)_3 \cdot 9H_2O$), iron chloride ($FeCl_3 \cdot 6H_2O$), aluminium chloride ($AlCl_3$) and potassium permanganate ($KMnO_4$) were purchased from Kelon Company (China). Iron sulfate ($Fe_2(SO_4)_3$) was purchased from Xilong Chemical Co., Ltd (China). All other chemicals were purchased from Kelon Company. All chemicals were of analytical grade. Ultrapure water was used for all the experiment.

## 2.2. Preparation of CNFs and modified CNFs

CNFs were prepared using the mechanical shearing method and were derived from wet bamboo pulp by employing the specific steps followed by Luo *et al.* [37]. The preparation conditions of modified CNFs were re-examined to determine the suitable modified CNFs for TCs removal. Specifically, the following preparation conditions were explored: (i) three metal salt modifiers ($AlCl_3$, $KMnO_4$ and $FeCl_3$), (ii) three iron salt modifiers ($FeCl_3$, $Fe(NO_3)_3$ and $Fe_2(SO_4)_3$), and (iii) various metal salt/CNFs mass ratios (0.5, 1, 1.5, 2, 2.5, 3 and 4).

Regarding the preparation of modified CNFs. First, the iron salt modifier was weighed and mixed with 1 wt% CNFs suspension in a certain mass ratio at room temperature, and the mixture was stirred for 24 h, and NaOH (1 M) was slowly added to the mixture during stirring. Second, deionized water was added to the mixed solution, and the solution was centrifuged at 8000 r.p.m. Then, the obtained modified material was washed with deionized water. Finally, the washed material was freeze-dried with a vacuum freeze dryer (LAB-1A-50E, China) for 48 h, and the dried modified CNFs were stored at room temperature. FeCl3-modified CNFs are hereinafter referred to as Fe(III)@CNFs.

## 2.3. Material characterization

The dried CNFs and Fe(III)@CNFs were coated with gold by using a vacuum sputter coater (Q150T ES, Quorum, UK). Subsequently, the morphological characteristics of CNFs and Fe(III)@CNFs were observed through scanning electron microscopy (SEM) (1530, LEO, Germany) at 15 kV. Energy-dispersive X-ray spectroscopy (EDS) was used to analyse element contents. CNFs and Fe(III)@CNFs were treated with vacuum heat for 4 h, and $N_2$ adsorption–desorption experiment was performed. The specific surface area and pore size analyser (V-Sorb 2800TP, Gold APP, China) and the Brunauer–Emmett–Teller

method were used to measure the specific surface area, pore volume and pore size. X-ray photoelectron spectroscopy (XPS) (ESCALAB 250Xi, Thermo Fisher Scientific, USA) was used to examine the binding energy under the Al K-alpha X-ray of a monochromator with an excitation source of 150 W. The XPS data were treated using XPSPEAK software for peak splitting. The surface functional groups of Fe(III)@CNFs and TCs-adsorbing Fe(III)@CNFs were measured using a Fourier-transform infrared (FTIR) spectrum analyser (Nicolet 6700, Nicolet, USA). The isoelectric point (IEP) of Fe(III)@CNFs was tested using a zeta potential/nanoparticle size analyser (Zetasizer Nano ZS90, Malvern, UK). The iron content in the leachate was tested and analysed by atomic absorption spectrophotometer (GGX-600, Haiguang Instrument Co., Ltd); specific parameters were designed as follows: air flow rate is $7.6\,l\,min^{-1}$, acetylene flow rate is $1.3\,l\,min^{-1}$, 370 V, 10 mA and measurement was performed at a wavelength of 248.3 nm. The thermal stability was tested by Thermogravimetric analysis-differential scanning calorimetry (TGA-DSC) (TGA/DSC3+/1600, Mettler Toledo) under 0–1000°C; specific parameters are designed as follows: nitrogen atmosphere, heating rate $20°C\,min^{-1}$. The crystalline phases in CNFs and Fe(III)@CNFs samples were tested and analysed by X-ray diffraction (XRD) (Empyrean, Panalytical); specific parameters are designed as follows: Cu target, 40 kV, 40 mA, within the $2\theta$ angle range of 3–90°, patterns were recorded at scan step size of 0.039 and time per step of 20.91. TGA-DSC and XRD analyses were detected by Analytical and Testing Center, Sichuan University, China.

## 2.4. Batch adsorption experiments

In the adsorption experiment, 50 mg of adsorbent was separated added to 100 ml of TC, CTC and OTC solutions ($10\,mg\,l^{-1}$). Subsequently, the working solutions were placed in a constant temperature shaking box (ZWY-2112B, Zhicheng, China) at 298 K and were shaken for 24 h in the absence of light until an equilibrium was attained. Finally, the supernatant was filtered using a 0.45 µm filter after precipitation and quantified using an ultraviolet–visible spectrophotometer (N4, INESA, China) at the wavelengths of 275 nm (for TC and OTC) and 227 nm (for CTC). The ultraviolet spectrum of TC, CTC and OTC was scanned in the wavelength range of 190–400 nm using ultraviolet–visible spectrophotometer (N4, INESA, China) and the wavelength corresponding to the strongest absorbance value was obtained. In addition, the concentration of TCs has an excellent linear relationship with absorbance at the corresponding wavelength. Unless otherwise stated, this regular adsorption experiment was completed at 298 K and pH 7.

To obtain the applicable pH range of adsorbents, the initial pH was adjusted from 3 to 12 (increment by 1) by using 0.1 M NaOH and HCl solutions. A digital multimeter (Multi 3420 set B, WTW, Germany) equipped with a pH electrode (SenTix®940–3, WTW, Germany) was used to assess the pH value. The impact of adsorbent dosage (10–100 mg) on TCs adsorption was investigated at optimal pH. Moreover, the influence of different concentrations of several anions ($Cl^-$, $CO_3^{2-}$ and $SO_4^{2-}$) on TCs adsorption was examined, and the adsorbent was added to the TCs solution with two coexisting anion concentrations (1 and 10 mM) at optimal pH. All the experiments were conducted in triplicate, and the mean values were used for the subsequent data analysis.

## 2.5. Kinetic and isotherm analysis of adsorption

To determine adsorption kinetic characteristics, sampling must be conducted at different time points (5, 10, 15, 30, 45, 60, 120, 240, 360, 480, 600 and 720 min) at optimal pH. The experimental conditions of adsorption isotherm features were mostly consistent with those for the determination of adsorption kinetic characteristics, except for the initial concentration of TCs ($4–500\,mg\,l^{-1}$), sampling points (after adsorption reached equilibrium) and temperature (15°C, 25°C and 35°C).

The obtained kinetics data were fitted to the pseudo-first order [48], pseudo-second order [49], Morris–Weber intra-particle diffusion [50] and simple Elovich [51] models. The data obtained at various temperatures were analysed using the Langmuir [52], Freundlich [53], Temkin [54] and Dubinin–Radushkevich (D–R) [55] models. The thermodynamic parameters of adsorption were calculated using thermodynamic equations [36]. The aforementioned linear equations of the kinetic, thermodynamics and isothermal models are presented in the electronic supplementary material.

## 2.6. Regeneration experiments

A total of 1 g of TCs-saturated Fe(III)@CNFs was placed in a 250 ml flask for regeneration experiments. Three regeneration methods were employed (NaOH solution, $H_2O_2 + UV$, and $NaOH + ultrasound$).

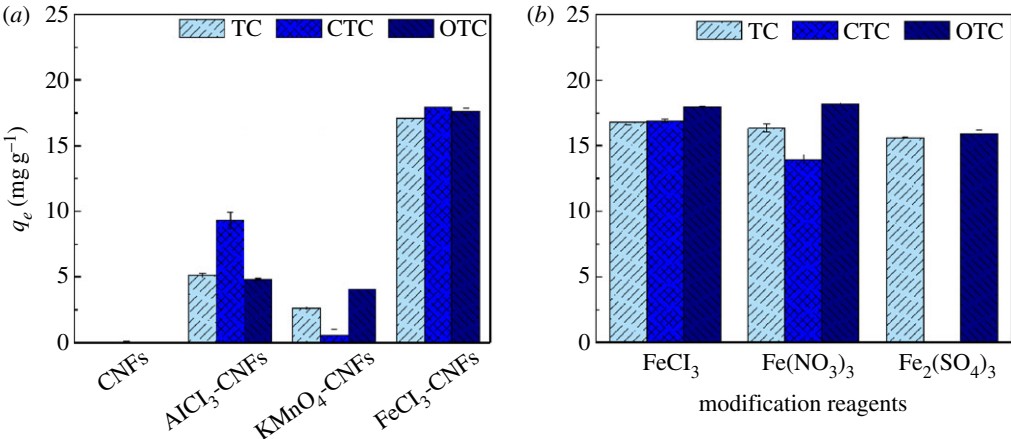

**Figure 1.** Effect of modification reagents on TCs adsorption capacity of modified CNFs: (*a*) metal salt and (*b*) iron salt (experimental condition: volume, 100 ml; adsorbent, 50 mg; initial concentration of TCs, 10 mg $l^{-1}$; initial pH, 7; *T*, 298 K).

In the first method, 100 ml of 1 mol $l^{-1}$ NaOH solution was added to the flask. In the second method, 100 ml of deionized water and 1 ml of $H_2O_2$ were added to the flask. Then, the flask was subjected to magnetic stirring, and the solution was desorbed for 2 h under UV light. In the third method, 100 ml of 1 mol $l^{-1}$ NaOH solution was added to the flask and sonicated for 2 h. After the selection of the optimal regeneration method, the effects of regeneration time on adsorption were investigated. The regenerated Fe(III)@CNFs were filtered and washed repeatedly with deionized water. The washed Fe(III)@CNFs were dried and re-used in the regular adsorption experiments. The regeneration–adsorption experiments were performed in five cycles with the optimal regeneration method.

Equations (2.1) and (2.2) present the TCs removal rate (RR%) and adsorption capacity ($q_e$) formulae

$$RR\% = \frac{C_0 - C_e}{C_0} \times 100 \tag{2.1}$$

and

$$q_e = \frac{C_0 - C_e}{m} \times V, \tag{2.2}$$

where RR% is the removal rate of TCs (%), $q_e$ is the adsorption capacity for TCs at equilibrium (mg $g^{-1}$), $C_0$ and $C_e$ are the initial and equilibrium concentrations, respectively, of TCs (mg $l^{-1}$), $V$ is the volume of TCs solutions (l) and $m$ is the mass of the adsorbent (g).

# 3. Results and discussion

## 3.1. Modification of CNFs

### 3.1.1. Different modifiers

To improve the TCs adsorption performance of the CNF adsorbent, aluminium, manganese and iron salts were used to modify CNFs. Figure 1*a* presents the TCs adsorption capacities of the CNFs and CNFs modified with different metal salts. CNFs exhibit almost no TCs adsorption. The low TCs adsorption capacities of natural CNFs are attributed to the small specific surface area and electrostatic repulsion [36]. The TCs adsorption capacities of CNFs modified with the iron salt were significantly higher than those of CNFs modified with the other two metal salts. The TCs adsorption capacities of CNFs modified with the iron salt were 3.3 times (TC), 1.9 times (CTC) and 3.7 (OTC) times higher than those of CNFs modified with the aluminium salt, and they were 6.5 times (TC), 31.7 times (CTC) and 4.3 times (OTC) higher than those of CNFs modified with the manganese salt. Alidadi *et al.* [56] also reported that Fe-modified sawdust exhibited the optimum TC removal efficiency among four modified agents. Thus, iron salt (III) may be a type of modifier that can be effectively used to improve the adsorption performance of CNFs.

(a) 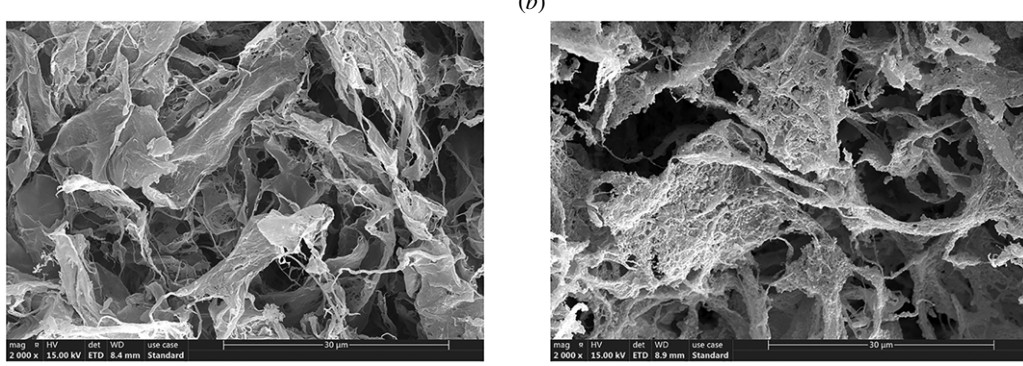 (b)

**Figure 2.** Typical scanning electron micrographs of (a) CNFs and (b) Fe(III)@CNFs.

Based on the TCs adsorption results of CNFs modified with different metal salts, the effects of different iron salt (III) modifiers were studied. $FeCl_3$, $Fe(NO_3)_3$ and $Fe_2(SO_4)_3$ were used to modify CNFs. The results are presented in figure 1b. The CNFs modified with $FeCl_3$ exhibited the optimum adsorption performance for all the three TCs. The adsorption capacities for TC, CTC and OTC were 16.81, 16.90 and 17.91 mg g$^{-1}$, respectively. Iron salt (III)-modified CNFs can be ranked in terms of their TCs adsorption capacities as follows: $FeCl_3$-modified CNFs > $Fe(NO_3)_3$-modified CNFs > $Fe_2(SO_4)_3$-modified CNFs. Therefore, the adsorption properties of only $FeCl_3$-modified CNFs, namely Fe(III)@CNFs, were investigated under various operational conditions in the following experiments. Iron-rich materials are effective adsorbents for TCs [47], but their application is limited by few active sites and small specific surface area. CNFs, as an iron load frame, can effectively overcome the disadvantages of high-reactivity iron materials. Thus, the $FeCl_3$-modified CNFs exhibit high TCs adsorption capacities.

### 3.1.2. Different Fe(III)/CNFs mass ratios

Fe(III)@CNFs-adsorbed TC, CTC and OTC showed the same trend when the Fe(III)/CNFs mass ratios varied from 0.5 to 4 (electronic supplementary material, figure S1). The adsorption capacity of Fe(III)@CNFs initially increased gradually with an increase in the Fe(III)/CNFs mass ratio and then reached the maximum point (18.32, 18.49 and 18.09 mg g$^{-1}$ for TC, CTC and OTC, respectively) when the Fe(III)/CNFs mass ratio was 2.5. The amount of Fe(III) loaded on the surface of Fe(III)@CNFs increased, thereby leading to an increase in the number of binding sites of adsorbents for TCs. However, when the Fe(III)/CNFs mass ratio increased further, the TCs adsorption amount became stable. Because the loading of Fe(III) on limited CNFs was saturated, the adsorption capacity of Fe(III)@CNFs did not further increase with an increase in the Fe(III)/CNFs mass ratio. Considering the adsorption capacities and chemical consumption, the Fe(III)/CNFs mass ratio of 2.5 was selected as the fixative value for the subsequent experiments.

## 3.2. Characterization of Fe(III)@CNFs

### 3.2.1. Surface morphology and physical parameters

The surface morphology and microstructure of CNFs and Fe(III)@CNFs were observed through SEM (figure 2). The surface of natural CNFs was smooth and dense and had no distinct porous structure (figure 2a). By contrast, the surface of Fe(III)@CNFs was rough and uneven and exhibited abundant dot-like holes (figure 2b). Compared with the SEM results of Fe(III)@CNFs obtained by Luo *et al.* [37], the Fe(III)@CNFs surface is highly porous due to the high Fe(III)/CNFs mass ratio, which proved that Fe(III) plays a crucial role in pore production. The SEM images of the samples suggest that Fe(III) loading on the CNFs surface led to an increase in the specific surface area and pore volume of CNFs after modification (electronic supplementary material, table S1). The specific surface area and pore volume of Fe(III)@CNFs (171 m$^2$ g$^{-1}$ and 0.180 cm$^3$ g$^{-1}$) were 6.8 and 4.5 times higher, respectively, than those of the original CNFs (25.3 m$^2$ g$^{-1}$ and 0.0400 cm$^3$ g$^{-1}$). The Barrett–Joyner–Halenda (BJH) average pore diameter of Fe(III)@CNFs was 5.44 nm, which was smaller than that of

the original CNFs. According to a study, the molecular diameter of TC is 1.27 nm [57]; thus, TC can permeate the pore interior of adsorbents with a sufficient pore size. The key factor affecting adsorption may be the relationship between Fe(III) and TCs.

### 3.2.2. Spectroscopic evidences for Fe(III) loading on CNFs

Electronic supplementary material, table S2 presents the elemental analysis results of EDS. The Fe content increased from 0 to 66.4% after CNFs modification, indicating that Fe was successfully loaded onto CNFs. The results are consistent with the characterization results obtained by Luo *et al.* [37]. However, Fe content in this study was higher than that of Luo *et al.* [37] because Fe(III)/CNFs mass ratio was higher.

The high-resolution XPS spectra illustrate various chemical bonds of Fe(III)@CNFs (electronic supplementary material, figure S2). The C 1s region was fitted into three peaks located at 284.44, 286.33 and 287.96 eV, which corresponded to the carbon–carbon (C–C)/carbon–hydrogen (C–H), carbon–carbon–oxygen (C–C–O) and carbon–oxygen (C=O)/oxygen–carbon–oxygen (O–C–O) bonds [58,59]. The spectrum of the O 1s region comprised two peaks, and their binding energies were 529.86 and 532.16 eV, corresponding to the Fe–O and Fe–OH bonds [47,60]. The peaks of the Fe 2p spectrum confirmed the existence of Fe in Fe(III)@CNFs, which is consistent with the results of the EDS elemental analysis. The spectrum of the Fe 2p region was fitted into three peaks, and their binding energies were 710.19, 712.11 and 724.06 eV, corresponding to Fe $2p_{3/2}$ and Fe $2p_{1/2}$ [60]. This finding implied that Fe mainly exists in the $Fe^{3+}$ form [36], which can be further proved using the FTIR spectra of Fe(III)@CNFs. Luo *et al.* [37] investigated the FTIR spectra of CNFs and Fe(III)@CNFs, and their results proved the existence of the iron–hydroxyl bond. The crystalline phases in CNFs and Fe(III)@CNFs were analysed by XRD (electronic supplementary material, figure S3). CNFs contain three peaks located at 15.6°, 22.6° and 34.5°, which belong to the typical spectrum of cellulose [61]. However, the peaks disappeared after Fe(III) loading, indicating the crystalline form of cellulose was destroyed after modification. Thus, the results of EDS, XPS and FTIR confirmed the successful loading of Fe(III) onto CNFs and indicated that Fe(III)@CNFs contain carboxyl groups, hydroxyl groups and Fe-(hydr)oxides.

TGA-DSC was conducted to examine the thermal stability of CNFs and Fe(III)@CNFs (electronic supplementary material, figure S4). For the TGA of the CNFs and Fe(III)@CNFs heated in the air, the weight loss below 250°C is attributed to the evaporation of adsorbed water [31]. A sharp weight loss (87.67%) of CNFs from 250° to 400°C and a gradual weight loss of Fe(III)@CNFs ranged 250–850°C were observed because of the decomposition of materials. At different temperatures, the weight loss of Fe(III)@CNFs was always less than that of CNFs, and the final weight of Fe(III)@CNFs can be maintained at about 50%, while the final weight of CNFs was as low as about 10%. The results of TGA-DSC indicated that the loading of Fe(III) improved the thermal stability of the CNFs.

## 3.3. Factors affecting the adsorption of TCs on Fe(III)@CNFs

### 3.3.1. Effect of initial pH

Figure 3 illustrates the effect of pH on TCs adsorption. In general, the adsorption capacity and removal rate of Fe(III)@CNFs for the three TCs showed a similar trend. At pH 4, the adsorption capacities reached the maximum values, which were 18.36, 18.26 and 18.76 mg g$^{-1}$ for TC, CTC and OTC, respectively. The adsorption capacity only slightly decreased, and the removal rate of TCs remained greater than 80% for a wide pH range. Specifically, the removal rates of TC, CTC and OTC were 84–93%, 81–91% and 81–92%, respectively, for the pH of 3–11, 4–8 and 3–9, respectively. The adsorption performance of TC, CTC and OTC sharply decreased when pH exceeded 11, 10 and 10, respectively. These results can be explained by the surface charge of the adsorbent and the influence of initial pH on the existing form of the adsorbate [57]. Further details are presented in §3.7. Although initial pH has an impact on TCs adsorption by Fe(III)@CNFs, Fe(III)@CNFs provide better TCs adsorption performance in a wide pH range than other adsorbents [59,62,63]. In a typical WWTP, the pH of influent is usually 5.5–8.0 [64]; thus, Fe(III)@CNFs have the potential to remove TCs from actual wastewater and save the cost of reagents required for adjusting pH.

In addition, the leaching content of iron was tested to verify the stability of the Fe(III)@CNFs. The experimental results showed that the iron content of Fe(III)@CNFs at pH 7.05 and 12.00 was less than 0.03 mg l$^{-1}$, while the iron content at pH 3.07 was 0.241 mg l$^{-1}$. Although Fe(III)@CNFs has iron

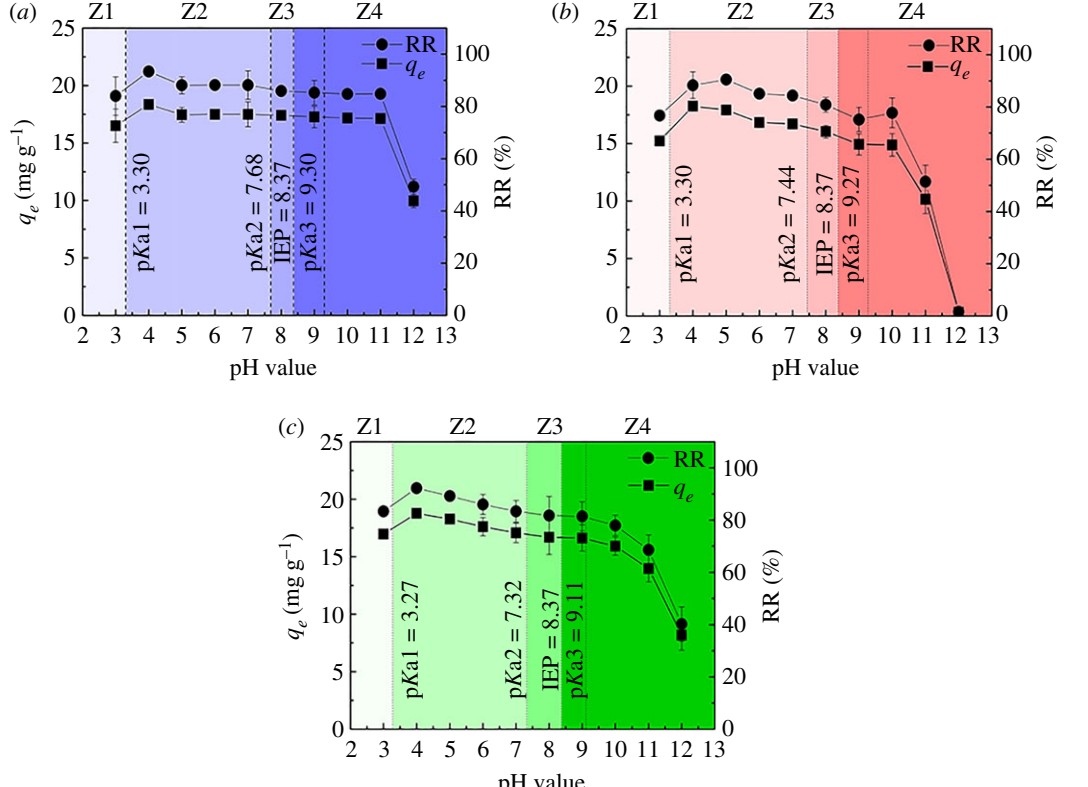

**Figure 3.** Effect of initial pH on TCs removal by Fe(III)@CNFs: (*a*) TC, (*b*) CTC and (*c*) OTC (experimental condition: volume, 100 ml; adsorbent, 50 mg; initial concentration of TCs, 10 mg l$^{-1}$; $T$, 298 K).

leaching at low pH, the leaching amount is very low. In addition, the adsorbent is mostly used in sewage with neutral pH, so the stability of the composites can be guaranteed.

### 3.3.2. Effect of Fe(III)@CNFs dosage

Because the adsorbent dose influences TCs adsorption performance by affecting the number of active sites, it must be optimized. Figure 4 illustrates the effect of the Fe(III)@CNFs dose on TCs removal.

The removal rate of TCs tends to increase with the dosage of Fe(III)@CNFs of 10–50 mg, and the removal rates of TC, CTC and OTC increased from 46.96%, 64.90% and 63.87% to 86.65%, 99.69% and 94.05%, respectively. After an adsorbent dosage of over 50 mg, the removal rate curve tended to become balanced with the increase in the adsorbent dosage. The rapid increase in removal rate in the initial stage was caused by a large increase in adsorption active sites. By contrast, when the dosage of Fe(III)@CNFs increased, the amount of TCs adsorbed per unit of the adsorbent sharply decreased. When the dosage of Fe(III)@CNFs was 50 mg, the adsorption capacities for TC, CTC and OTC were 17.29, 19.51 and 18.96 mg g$^{-1}$, respectively. The higher was the dosage of the adsorbent, the higher was the unsaturation of the surface active sites of the unit adsorbent. Similar trends were observed in other studies [65,66]. In view of the optimal removal rate of TCs and the economy of adsorbent dosage under experimental conditions, the optimal dosage of Fe(III)@CNFs was 50 mg.

### 3.3.3. Effect of coexisting anions

Coexisting ions affect the adsorption of organic matter through the 'salting out' and 'squeeze out' effects [57]. Actual wastewater has different concentrations and types of coexisting ions, and their presence may interfere with TCs adsorption. TCs mainly exist in the form of anionic species at neutral pH in actual wastewater (see §3.7). Hence, the influence of Cl$^-$, CO$_3^{2-}$ and SO$_4^{2-}$ coexisting anions on adsorption was inspected (figure 5).

For TC, low concentrations of CO$_3^{2-}$ and SO$_4^{2-}$ led to an increase in the adsorption capacities, and high concentrations of CO$_3^{2-}$ and SO$_4^{2-}$ resulted in a decrease in the adsorption capacities. For CTC, CO$_3^{2-}$ led

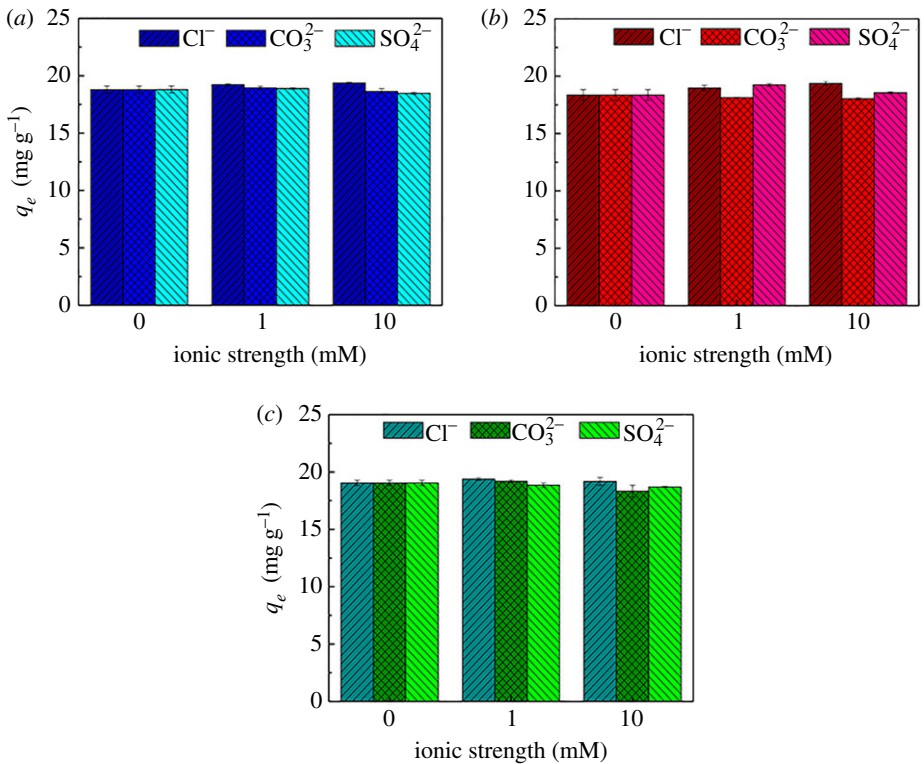

**Figure 4.** Effect of adsorbent dosage on TCs removal by Fe(III)@CNFs (*a*) TC, (*b*) CTC and (*c*) OTC (experimental condition: volume, 100 ml; initial pH, 4; initial concentration of TCs, 10 mg l$^{-1}$; *T*, 298 K).

**Figure 5.** Effect of ionic types and strength on TCs removal by Fe(III)@CNFs (*a*) TC, (*b*) CTC and (*c*) OTC (experimental condition: volume, 100 ml; initial pH, 4; initial concentration of TCs, 10 mg l$^{-1}$; *T*, 298 K; adsorbent, 50 mg).

to a decrease in the adsorption capacities, and $SO_4^{2-}$ resulted in an increase in the adsorption capacities, regardless of low and high ion concentrations. For OTC, the effect of different concentrations of $CO_3^{2-}$ was consistent with its effect on TC, and $SO_4^{2-}$ led to a decrease in the adsorption capacities regardless of low and high ion concentrations. Additionally, at the low concentration of $Cl^-$ (1 mM), the adsorption

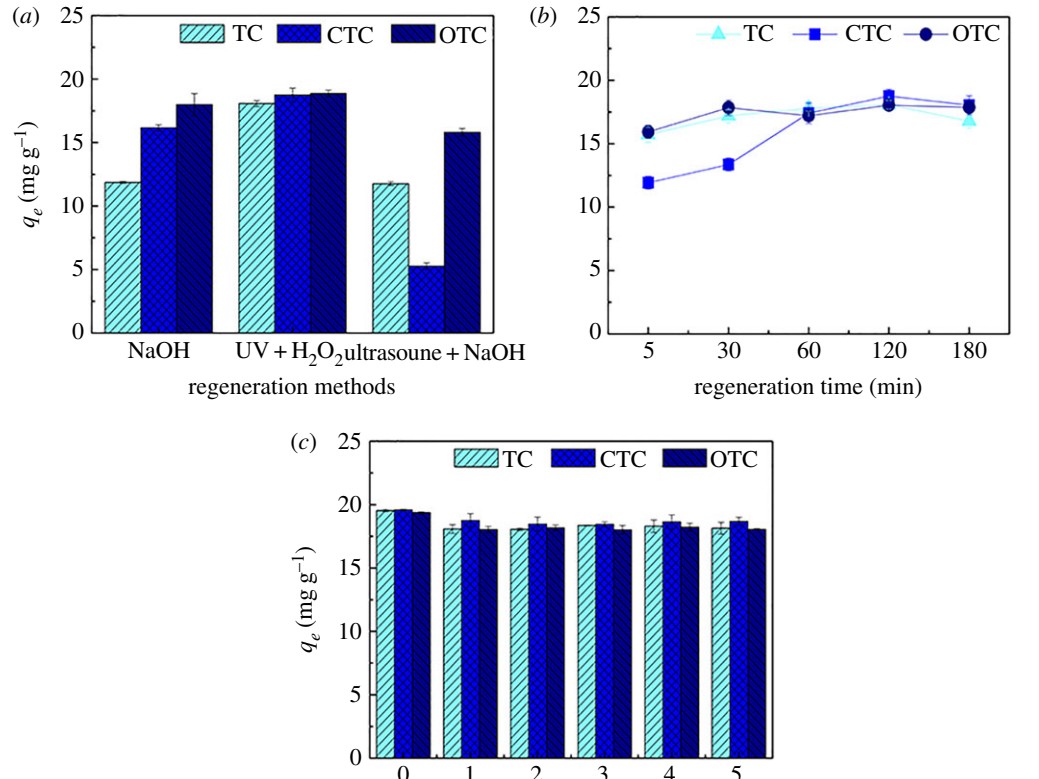

**Figure 6.** Effect of regeneration on TCs removal by Fe(III)@CNFs (*a*) regeneration methods, (*b*) regeneration time and (*c*) regeneration cycles (experimental condition: volume, 100 ml; initial pH, 4; initial concentration of TCs, 10 mg l$^{-1}$; *T*, 298 K; adsorbent, 50 mg).

capacities for TC, CTC and OTC increased by 2.25%, 3.40% and 1.80%, respectively; and in the presence of high concentration of Cl$^-$ (10 mM), the adsorption capacities for TC, CTC and OTC increased by 3.00%, 5.56% and 0.67%, respectively.

In general, under different ion types and concentrations, the positive effects of Cl$^-$, CO$_3^{2-}$ and SO$_4^{2-}$ ions were below 5.56%, and the negative impact was below 3.71%. The results indicated that the influences of the three coexisting ions on the TCs adsorption process were insignificant, and the inner spherical complex was barely affected by coexisting ions [67]. Zhang *et al.* [59] found that TCs adsorption by the amino-iron(III)-functionalized SBA15 was not affected by NaCl because of the formation of an inner spherical complex. Gu & Karthikeyan [68] reached the same conclusion as Zhang *et al.* [59] after studying the effect of coexisting ions on TC adsorption by Fe and Al hydrated oxides. Thus, a complex reaction occurred between TCs and Fe(III)@CNFs to form the inner spherical complex.

## 3.4. Fe(III)@CNFs regeneration performance

Adsorbent reusability is highly significant for large-scale commercial applications. The commonly used regeneration methods include pyrolysis, gasification, chemical solvents, microorganisms, electrochemistry, ultrasound and wet air oxidation [69]. Conspicuous differences existed among the three regeneration methods used to regenerate and re-use Fe(III)@CNFs to adsorb TCs (figure 6*a*). The adsorption capacities of Fe(III)@CNFs regenerated using NaOH and ultrasound + NaOH considerably decreased. However, Fe(III)@CNFs regenerated using UV + H$_2$O$_2$ maintained high adsorption capacities for the three TCs (18.08, 18.75 and 18.87 mg g$^{-1}$ for TC, CTC and OTC, respectively). The regeneration performance of UV + H$_2$O$_2$ is excellent due to three reasons. First, the hydroxyl, carboxyl and carbonyl groups of Fe(III)@CNFs are protonated under the acidic condition of pH 4, and protons can easily replace bonded TC ions [38]. Second, UV + H$_2$O$_2$ generates hydroxyl radicals with strong oxidizing ability to destroy the bridge between TCs and adsorbent surface functional groups through oxidation and desorption, and thus, Fe(III)@CNFs return to the original state [70]. Finally, the desorbed TCs are also oxidized by hydroxyl radicals, which lowers the possibility of TCs re-

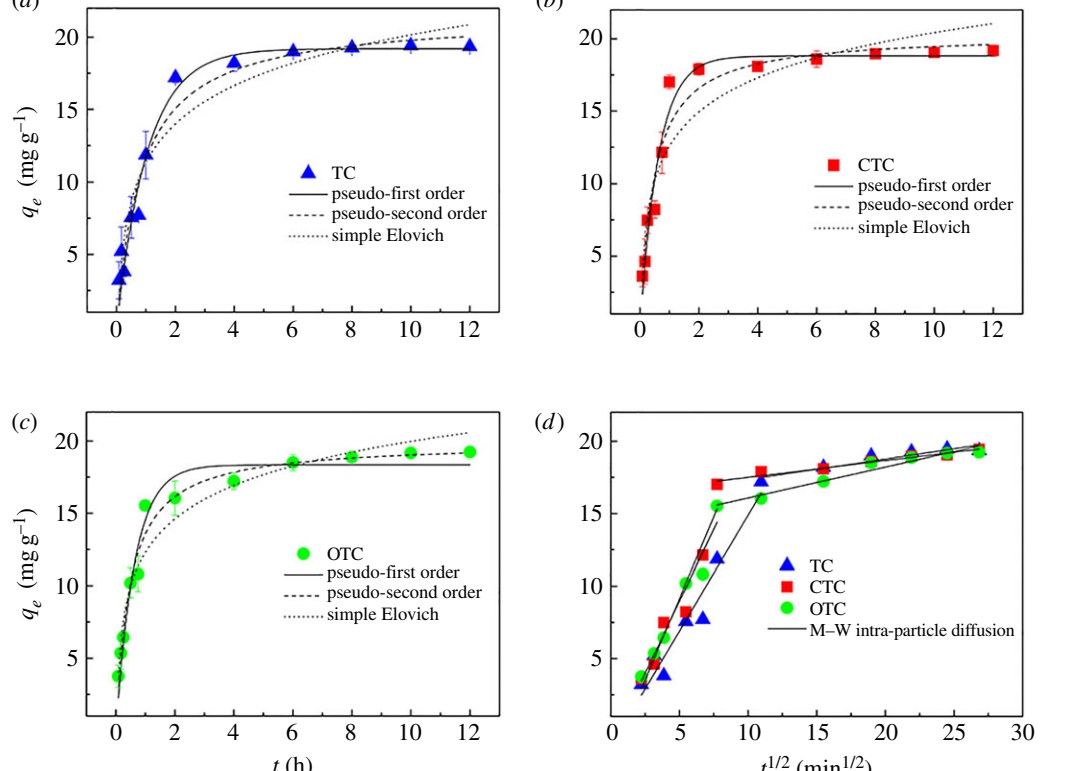

**Figure 7.** Adsorption kinetics of TCs on Fe(III)@CNFs (experimental condition: volume, 100 ml; initial pH, 4; initial concentration of TCs, 10 mg l$^{-1}$; $T$, 298 K; adsorbent, 50 mg).

adsorption onto Fe(III)@CNFs, thereby leading to stable adsorption performance. However, the two commonly used regeneration methods, NaOH and ultrasound + NaOH, led to a considerable decrease in the adsorption capacities of Fe(III)@CNFs. The carboxyl groups of Fe(III)@CNFs were deprotonated due to the NaOH solution, leading to an increase in the pH value, which is not conducive to TCs desorption [38]. In addition, ultrasound is mainly effective for physically adsorbed adsorbates. Considering the effectiveness of regeneration, $UV + H_2O_2$ was selected as the optimum regeneration method for Fe(III)@CNFs.

The determination of an appropriate regeneration time leads to not only the restoration of the original state of the adsorbent to the highest extent but also the improvement of the efficiency of the actual engineering treatment. Figure 6b shows that the adsorption capacities of Fe(III)@CNFs increased with an increase in the regeneration time. Maximum adsorption capacities were attained when the regeneration time was 2 h, and then, the adsorption capacity tended to become stable. The regeneration time affects TCs oxidation and desorption. When the regeneration time was 2 h, $UV + H_2O_2$ could perfectly restore the adsorption performance of Fe(III)@CNFs. Consequently, the $UV + H_2O_2$ regeneration treatment time was set to 2 h.

After repeated regeneration, an adsorbent that can maintain stable adsorption performance is required for practical applications. Five regeneration–adsorption cycles were conducted using $UV + H_2O_2$ regeneration (figure 6c). Compared with the TCs adsorption capacities of original Fe(III)@CNFs, those of Fe(III)@CNFs after five cycles remained greater than 92%. The results of regeneration performance showed that Fe(III)@CNFs can be effectively regenerated and re-used, and thus, they can be widely used for removing TCs from actual wastewater.

## 3.5. Adsorption kinetics

The exploration of adsorption kinetics is a crucial part of reflecting the rate change of the adsorption process and providing the potential adsorption mechanism [71]. Figure 7 shows the trend of TCs adsorption capacities of Fe(III)@CNFs with the contact time. Fe(III)@CNFs-adsorbed TCs at the highest speed in the first 1 h, and the adsorption capacities for TC, CTC and OTC accounted for

**Table 1.** Parameters for the adsorption kinetics models of TCs on Fe(III)@CNFs.

| kinetics models | tetracycline antibiotics | | |
| --- | --- | --- | --- |
| | TC | CTC | OTC |
| pseudo-first order | | | |
| $q_e$ (mg g$^{-1}$) | 19.20 | 18.82 | 18.35 |
| $k_1$ (h$^{-1}$) | 0.949 | 1.603 | 1.576 |
| $R^2$ | 0.968 | 0.964 | 0.960 |
| pseudo-second order | | | |
| $q_e$ (mg g$^{-1}$) | 21.47 | 20.35 | 19.92 |
| $k_2$ (g mg$^{-1}$ h$^{-1}$) | 0.055 | 0.108 | 0.107 |
| $R^2$ | 0.996 | 0.999 | 0.999 |
| M–W intra-particle diffusion | | | |
| $k_{pi}$ (mg g$^{-1}$ min$^{-1/2}$) | 0.70 | 0.59 | 0.59 |
| $k_{pi1}$ (mg g$^{-1}$ min$^{-1/2}$) | 1.60 | 2.25 | 1.99 |
| $k_{pi2}$ (mg g$^{-1}$ min$^{-1/2}$) | 0.14 | 0.12 | 0.21 |
| $C_{pi}$ | 4.06 | 6.51 | 6.20 |
| $R_1^2$ | 0.932 | 0.911 | 0.949 |
| simple Elovich | | | |
| $b$ | 3.83 | 3.41 | 3.33 |
| $a$ | 11.33 | 12.58 | 12.29 |
| $R^2$ | 0.931 | 0.889 | 0.947 |

61.25%, 88.67% and 80.73%, respectively, of the final equilibrium adsorption capacities. Such a high adsorption rate is attributed to the high concentration of TCs in the initial stage of adsorption, and hence, a large concentration gradient is conducive to the mass transfer resistance between water and the adsorbent surface [62]. In addition, the adsorbent can provide sufficient effective active sites in the initial stage. After 1 h, the adsorption rate gradually decreased until it reached the equilibrium at 8 h. From the middle to late stage of adsorption, the driving force between the solution and Fe(III)@CNFs is insufficient. Moreover, most reactive sites are occupied by adsorbates, and electrostatic repulsion occurs between the absorbed TCs and TCs present in the solution. Hence, adsorption tends to balance.

The obtained kinetic experimental data were fitted to the pseudo-first-order kinetic, pseudo-second-order kinetic, Morris–Weber intra-particle diffusion and simple Elovich models (figure 7). Table 1 presents the kinetic parameters of TCs adsorption on Fe(III)@CNFs and the correlation coefficient ($R^2$). On comparing $R^2$, the pseudo-second-order kinetic model ($R^2 > 0.99$) was found to be more apt for describing TCs adsorption on Fe(III)@CNFs than the pseudo-first-order kinetic model ($R^2$: 0.960, 0.964 and 0.968). Thus, the pseudo-second-order kinetic model represents the adsorption process of TCs on Fe(III)@CNFs, which is consistent with the findings of many TCs studies [59,60,62,72,73]. The adsorption of TCs on Fe(III)@CNFs is controlled with a chemical process, and the adsorption rate is proportional to the number of surface active sites [72]. In addition, the adsorption rate constants ($k_2$) of TC, CTC and OTC were compared, and their order was as follows: CTC > OTC > TC, which indicated that Fe(III)@CNFs can adsorb CTC fastest.

The Morris–Weber intra-particle diffusion model was employed to explore the actual rate-limiting step [57] (figure 7d). Before the adsorption equilibrium, $q_t$ and $t^{1/2}$ presented two linearities. In the rapid adsorption stage, the TCs were stacked on the outer surface of Fe(III)@CNFs, and the rate-limiting step was liquid film diffusion. In the slow adsorption stage, the rate-limiting step was intra-particle diffusion. In addition, intra-particle diffusion can be only considered the rate control step if the fitting straight line passes through the origin [50]. However, the result indicated that intra-particle diffusion is not the only control step for TCs adsorption since the line did not go through the origin. $C_{pi} > 0$ of the Morris–Weber intra-particle diffusion model manifests that TCs are rapidly adsorbed onto Fe(III)@CNFs [74]. The simple Elovich model ($R^2$: 0.931, 0.889 and 0.947) can also describe

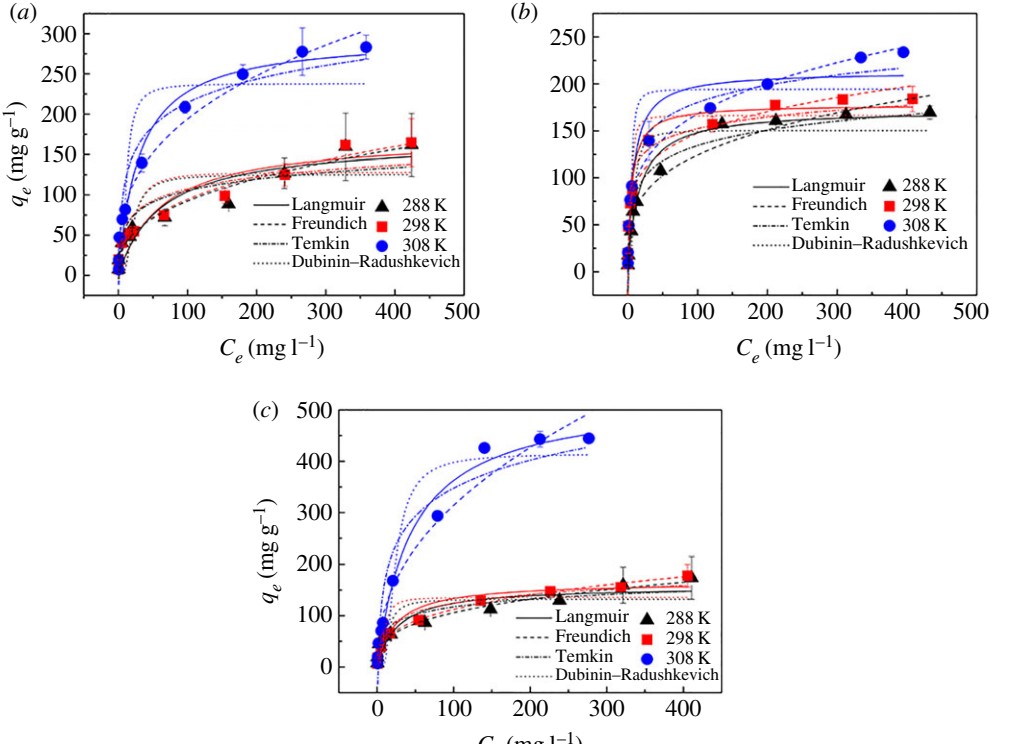

**Figure 8.** Equilibrium adsorption isotherms of TCs on Fe(III)@CNFs (a) TC, (b) CTC and (c) OTC (experimental condition: volume, 100 ml; initial pH, 4; initial concentration of TCs, 4–500 mg l$^{-1}$; T, 288, 298, 308 K; adsorbent, 50 mg).

adsorption. This model assumes that the surface adsorption energy is uniformly distributed and that there is no desorption [57,75].

## 3.6. Adsorption isotherms and thermodynamics

Figure 8 shows the adsorption isotherms describing the equilibrium relationship between Fe(III)@CNFs and TCs. The adsorption capacities of Fe(III)@CNFs increase with an increase in the initial concentration of TCs, and then tend to become stable. A higher initial concentration of TCs can provide a stronger concentration driving force to overcome the resistance between solids and liquids. If the quantitative adsorbents interact with TCs in the solution to reach adsorption saturation, the adsorption capacities can no longer change with an increase in TCs. In addition, the adsorption capacities of Fe(III)@CNFs are proportional to temperature, indicating that the adsorption of TCs on Fe(III)@CNFs is endothermic [62].

The experimental data were applied to four adsorption isotherm models: Langmuir, Freundlich, Temkin and D–R models. Figure 8 presents the fitting curve. Table 2 presents the isotherm parameters and $R^2$. On comparing $R^2$, the Langmuir model was found to be more appropriate for describing TCs adsorption because the $R^2$ of the Langmuir model (varying between 0.917 and 0.998) is higher than that of other models under most conditions. Compared with the D–R model, the $R^2$ of Temkin was higher, which indicated that electrostatic attraction and heterogeneous pores caused the exothermic reaction [57]. Hence, TCs adsorption on Fe(III)@CNFs is dominated by monolayer molecular adsorption, and Fe(III)@CNFs have a homogeneous surface [72]. At 308 K, the maximum adsorption capacities ($q_m$) of Fe(III)@CNFs for TC, CTC and OTC calculated using the Langmuir model were 294.12, 232.56 and 500.00 mg g$^{-1}$, respectively. The difference of chemical structure among OTC, TC and CTC is the substituent. The adding amount of hydroxyl in OTC resulted in the increase of combination sites with Fe(III)@CNFs. Fe(III)@CNFs contain carboxyl groups and hydroxyl functional groups that can lead to the formation of a hydrogen bond with hydroxyl of OTC [57]. Thus, the adsorption capacities of Fe(III)@CNFs to OTC are higher than adsorption capacity to TC. In addition, the additional chlorine atom at C7 position in CTC have the potential effect on electric density of the whole molecules and enhanced polarity of CTC's functional groups [76]; therefore, the adsorption capacity of Fe(III)@CNFs to CTC is just 232.56 mg g$^{-1}$. Comparing with CTC, Fe(III)@CNFs can form the more complex with OTC and TC.

**Table 2.** Isotherm parameters of TCs adsorption on Fe(III)@CNFs.

| isotherm models | TC | | | CTC | | | OTC | | |
|---|---|---|---|---|---|---|---|---|---|
| | 288 K | 298 K | 308 K | 288 K | 298 K | 308 K | 288 K | 298 K | 308 K |
| **Langmuir** | | | | | | | | | |
| $q_m$ (mg g$^{-1}$) | 163.93 | 166.67 | 294.12 | 175.44 | 185.19 | 232.56 | 169.49 | 175.44 | 500.00 |
| $K_L$ (l mg$^{-1}$) | 0.02 | 0.03 | 0.04 | 0.07 | 0.16 | 0.08 | 0.03 | 0.04 | 0.03 |
| $R^2$ | 0.917 | 0.938 | 0.991 | 0.998 | 0.998 | 0.990 | 0.959 | 0.959 | 0.982 |
| **Freundlich** | | | | | | | | | |
| $K_F$ (mg$^{(1-n)}$ l$^n$ g$^{-1}$) | 16.22 | 16.44 | 22.68 | 20.29 | 41.89 | 36.32 | 19.00 | 15.96 | 19.67 |
| $1/n$ | 0.38 | 0.38 | 0.47 | 0.40 | 0.28 | 0.34 | 0.38 | 0.43 | 0.61 |
| $R^2$ | 0.931 | 0.939 | 0.909 | 0.948 | 0.965 | 0.938 | 0.893 | 0.903 | 0.921 |
| **Temkin** | | | | | | | | | |
| $b$ (J mol$^{-1}$) | 116.40 | 117.93 | 60.69 | 101.54 | 138.80 | 96.00 | 109.88 | 99.13 | 32.55 |
| $K_T$ (l g$^{-1}$) | 1.67 | 1.67 | 1.60 | 2.99 | 50.95 | 8.59 | 2.16 | 1.44 | 0.83 |
| $R^2$ | 0.838 | 0.855 | 0.953 | 0.946 | 0.925 | 0.966 | 0.920 | 0.962 | 0.930 |
| **D–R** | | | | | | | | | |
| $q_s$ (mg g$^{-1}$) | 125.40 | 128.03 | 237.89 | 150.43 | 166.40 | 194.35 | 132.09 | 135.67 | 414.90 |
| $K_{DR}$ (mol$^2$ kJ$^{-2}$) | 57.66 | 56.68 | 15.91 | 10.67 | 3.11 | 3.08 | 25.50 | 11.30 | 66.13 |
| $E$ (kJ mol$^{-1}$) | −0.09 | −0.09 | −0.18 | −0.22 | −0.40 | −0.40 | −0.14 | −0.21 | −0.09 |
| $R^2$ | 0.675 | 0.687 | 0.829 | 0.867 | 0.855 | 0.829 | 0.692 | 0.766 | 0.903 |

**Table 3.** Comparison of Langmuir maximum adsorption capacity and experimental parameters of various adsorbents for TCs removal.

| original material | modifier | adsorbate | reaction condition (pH; $T$; $C_0$) | $q_m$ (mg g$^{-1}$) | reference |
|---|---|---|---|---|---|
| cellulose nanofibres | FeCl$_3$ | TC | 4; 308 K; <500 mg l$^{-1}$ | 294.12 | this study |
| | | CTC | | 232.56 | |
| | | OTC | | 500.00 | |
| nanocrystalline cellulose | NA[a] | TC | 5; 318 K; NA | 13.253 | [38] |
| hydrous ferric oxide | NA | TC | NA; 318 K; <70 mg l$^{-1}$ | 99.49 | [60] |
| granular activated carbon | NA | TC | 3; 298 K; <25 mg l$^{-1}$ | 85.29 | [77] |
| cellulose nanofibril | graphene oxide | TC | NA; 298 K; NA | 454.6 | [46] |
| | | | | 478.9 | |
| | | CTC | | 486.7 | |
| | | OTC | | | |
| sawdust | Fe(OH)$_3$ | TC | 7.8; 295 K; <30 mg l$^{-1}$ | 5.41 | [56] |
| magnetic microsphere | Fe$^{2+}$ | TC | NA; 303 K; <1000 mg l$^{-1}$ | 166 | [70] |
| bovine serum albumin | Fe$_3$O$_4$ | TC | NA; 298 K; <200 mg l$^{-1}$ | 104.35 | [78] |
| mesoporous silica SBA15 | amino-Fe(III) | TC | 5; 318 K, <95.78 mg l$^{-1}$ | 68.63 | [59] |
| | | CTC | | 43.07 | |
| | | OTC | | 31.84 | |

[a]NA: not available.

**Table 4.** The thermodynamic parameters for the adsorption of TCs on Fe(III)@CNFs.

| TCs | $T$ (K) | $\Delta G^0$ (kJ mol$^{-1}$) | $\Delta S^0$ (J mol$^{-1}$ K) | $\Delta H^0$ (kJ mol$^{-1}$) |
|---|---|---|---|---|
| tetracycline (TC) | 288 | −7.94 | 42.45 | 4.30 |
| | 298 | −8.31 | | |
| | 308 | −8.79 | | |
| chlortetracycline (CTC) | 288 | −6.89 | 282.10 | 73.63 |
| | 298 | −11.99 | | |
| | 308 | −12.43 | | |
| oxytetracycline (OTC) | 288 | −6.95 | 53.35 | 8.48 |
| | 298 | −7.29 | | |
| | 308 | −8.025 | | |

Table 3 presents the summary and comparison of the maximum adsorption capacity of the adsorbents used to remove TCs. The adsorption capacities of Fe(III)@CNFs are better than those of most other adsorbents reported in the literature. Although the nanocellulose-mixed graphite oxide aerogel prepared by Yao *et al.* [46] has higher adsorption capacities, its preparation difficulty and cost are higher than those of the proposed Fe(III)@CNFs. The aforementioned results confirmed that efficient Fe(III)@CNFs have the characteristics of low production cost, wide pH application range, and good regeneration performance and can be prepared using a simple method. Therefore, Fe(III)@CNFs are TCs adsorbents with excellent adsorption performance and a high practical application value.

To further study the impact of temperature on TCs adsorption, the relevant thermodynamic parameters, such as Gibbs free energy ($\Delta G^0$), entropy ($\Delta S^0$) and enthalpy ($\Delta H^0$), were calculated from the thermodynamic equation. Table 4 presents these parameters. The $\Delta G^0$ of the three TCs was less than 0 and decreased with an increase in temperature. Thus, adsorption is spontaneous [59], and higher

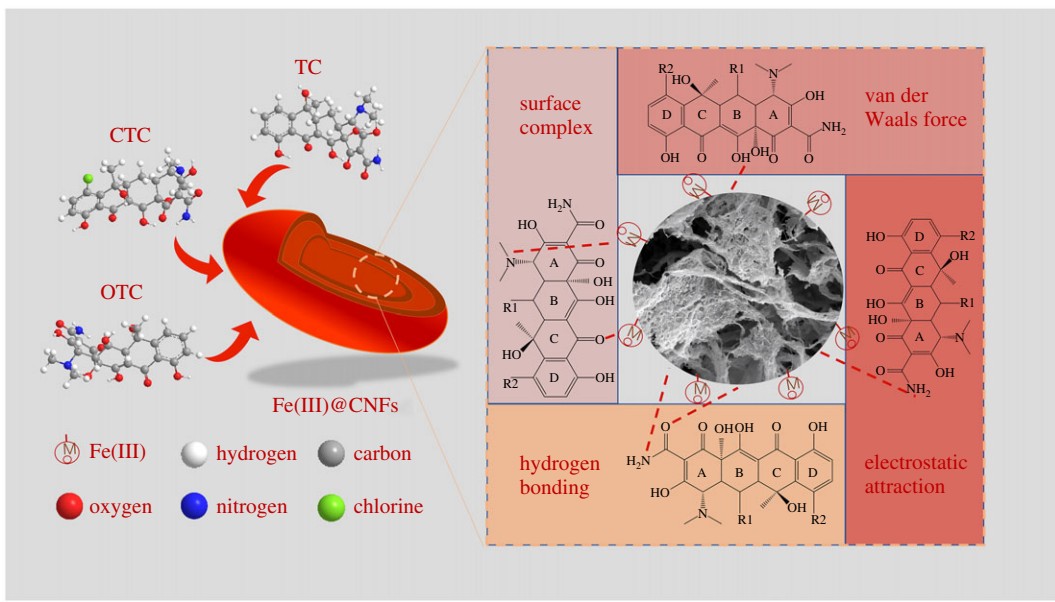

**Figure 9.** Potential mechanisms of Fe(III)@CNFs adsorbing TCs.

temperature is beneficial for TCs adsorption by Fe(III)@CNFs. $\Delta S^0$ of greater than 0 indicates that the surface structure of the adsorbents changes and that the disorder of the adsorption process increases, which is favourable for adsorption [59,60]. $\Delta H^0$ of greater than 0 indicates not only that adsorption of TCs on Fe(III)@CNFs is endothermic but also that this adsorption is a chemical process [59].

## 3.7. Mechanism analysis

The adsorption of TCs on Fe(III)@CNFs is more in line with the chemical process represented by the pseudo-second-order kinetic model. Surface complexation always dominates adsorption in each situation between Fe(III)@CNFs and TCs. The hydrogen bond and electrostatic interactions play a secondary role in adsorption. In addition, van der Waals forces participate in this process [57]. Figure 9 presents the foremost adsorption mechanisms.

The surface complex reaction between Fe(III) and TCs is of considerable significance for TCs adsorption onto Fe(III)@CNFs. Compared with the original CNFs, the loading of Fe(III) on the CNFs framework significantly improved the adsorption of TCs (figure 1). TCs have electron-rich ketones, carboxyl groups, amino groups and hydroxyl groups, which are conducive to the complex reaction between TCs and metals. In Chen and Huang's study [76], TC, CTC and OTC adsorb strongly to aluminium oxide ($Al_2O_3$). Adsorption sites were also simulated force fields, which protonated carbonyl and amino groups of TCs that can bind cations effectively and form complexes [79]. Fe(III) tends to interact with tricarbonylamide and oxygen present in the C ring of TCs to form inner spherical complexes [80]. The Fe(III) ions with TC, CTC and OTC form the stable complexes with the phenolic β-diketone structure of TCs [30]. In our study, the presence and absence of Fe(III) have a significant effect on the adsorption of TCs, thus the surface complexation between Fe(III) and TCs can be verified.

The functional groups of Fe(III)@CNFs can contribute to the adsorption of TCs by forming hydrogen bonds. Fe(III)@CNFs contain carboxyl and hydroxyl functional groups (see §3.2.2) that can lead to hydrogen bond formation with phenolic hydroxyl, amine, hydroxyl and ketene moieties of TCs [57]. In addition, the FTIR spectra of Fe(III)@CNFs before and after TCs adsorption were investigated (electronic supplementary material, figure S5). After Fe(III)@CNFs-adsorbed TCs, the peak appearing at $1636\ cm^{-1}$ moved to 1632, 1604 and $1623\ cm^{-1}$, and the peak observed at $1545\ cm^{-1}$ shifted to 1540, 1498 and $1517\ cm^{-1}$. The results indicated that multiple groups such as hydroxyl were involved in the removal of TCs present in the solution. The vibration peak of −OH change because of hydrogen bonding that restricts the vibration of −OH in Fe(III)@CNFs [80].

The surface charge of the adsorbent and the influence of initial pH on the existing form of the adsorbate can be used to explore electrostatic interactions. The pH corresponding to the IEP of Fe(III)@CNFs was 8.37 (electronic supplementary material, figure S6). TCs species are characterized by three acid dissociation constants (pKas) that delimit cationic (TCs⁺), zwitterionic (TCs±) and anionic

(TCs⁻ and TCs⁻ ⁻) species. The p$K$a values of TC are 3.30, 7.68 and 9.30, those of CTC are 3.30, 7.44 and 9.27, and those of OTC are 3.27, 7.32 and 9.11 [59]. Therefore, the pH of 3–12 can be divided into four zones (Z1, Z2, Z3 and Z4) in terms of IEP, p$K$a1 and p$K$a2 values (figure 3). For Z1 (pH < p$K$a1), TCs⁺ and Fe(III)@CNFs both had positive charges, which led to electrostatic repulsion. For Z1 to Z2 (p$K$a1 < pH < p$K$a2), TCs± caused by the deprotonation of the tricarbonylamide group was the dominant species, and electrostatic repulsion weakened, thereby highly enhancing the adsorption capacities for TCs. In this stage (Z1 to Z2), the electrostatic condition was unfavourable, and the adsorption capacities for TCs were mainly maintained through surface complexation [68,81]. For Z2–Z3 (p$K$a2 < pH < IEP), the dominant TCs species changed from TCs± to TCs⁻ due to the deprotonation of the phenolic diketone group of TCs and electrostatic attraction between Fe(III)@CNFs and TCs. However, the adsorption capacity slightly decreased in this situation, and the effect of electrostatic attraction was indistinct. For Z3–Z4 (pH > IEP), when the alkalinity of the solutions increased, the negatively charged Fe(III)@CNFs and anionic species (TCs⁻ and TCs⁻ ⁻) led to strong electrostatic repulsion. Furthermore, immoderate OH⁻ in the solution competed with TCs⁻ for active sites on Fe(III)@CNFs. The variations in the adsorption capacities and removal rates of TC, CTC and OTC at different pH values may be attributed to different p$K$a values and chemical groups [48,52]. The electrostatic interaction occupies a significant position in acidic and alkaline pH.

# 4. Conclusion

In this study, a renewable and non-sintered adsorbent with Fe(III) incorporated into CNFs was prepared and applied to adsorb TCs (TC, CTC and OTC) from an aqueous solution. After comparing the impact of various modifiers and Fe(III)/CNFs mass ratios on TCs adsorption, the preparation parameters of Fe(III)@CNFs suitable for three TCs adsorption simultaneously were obtained. The adsorbents showed a clear and rough pore structure, and the physical parameters changed after Fe(III) modification. Spectroscopic evidence indicated that Fe(III)@CNFs have multiple oxygen-containing functional groups and Fe(III) metal sites. The batch experiments with different pH values indicated that the manufactured Fe(III)@CNFs exhibited stable adsorption and excellent tolerance in a wide pH range that tend to reduce the cost of pH adjustment. Different types and concentrations of coexisting ions have various effects on TCs adsorption, and these effects are relatively weak. UV + H₂O₂ with the optimal regeneration performance was selected to regenerate Fe(III)@CNFs, and the original TCs adsorption capacities of Fe(III)@CNFs remained stable (greater than 92%) after five cycles. The characteristics of not being easily affected by coexisting ions, the strong reproducibility and firm Fe(III) greatly broaden the application possibilities of Fe(III)@CNFs in actual sewage treatment. The fitting data of kinetic, isotherm and thermodynamic proved that the spontaneous and endothermic adsorption mainly resulted from monolayer chemical adsorption, and the maximum adsorption capacities of Fe(III)@CNFs for TC, CTC and OTC were 294.12, 232.56 and 500.00 mg g⁻¹, respectively. The potential adsorption mechanisms of TCs onto Fe(III)@CNFs involved inner spherical surface complexation, hydrogen bonding, electrostatic interaction and van der Waals forces between the functional groups of TCs and Fe(III)@CNFs. The cost-effective Fe(III)@CNF adsorbents with excellent adsorption performance in a wide pH range and outstanding regeneration performance can be used to remove TCs from wastewater.

Data accessibility. The data are provided in electronic supplementary material [82].

Authors' contributions. L.L.: writing—original draft, experiment, data curation. M.L.: project administration, conceptualization, funding acquisition, writing—review and editing. Y.C.: conceptualization, supervision, writing—review and editing, methodology. Y.L.: experiment.

Competing interests. We declare we have no competing interests.

Funding. This work was financially supported by the Science and Technology Major Projects of Sichuan Province 'Technology Integration and Demonstration of Stability and Standard Achievement in Urban Sewage Treatment Plant' (no. 2019YFS0501).

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
