## [Peer Review File · Royal Society Open Science]

Review History

RSOS-210336.R0 (Original submission)

Review form: Reviewer 1

Is the manuscript scientifically sound in its present form?

Yes

Are the interpretations and conclusions justified by the results?

Yes

Is the language acceptable?

Yes

Do you have any ethical concerns with this paper?

No

Have you any concerns about statistical analyses in this paper?

No

Recommendation?

Major revision is needed (please make suggestions in comments)

Comments to the Author(s)

In this manuscript, the authors prepared Fe(III)@CNFs and characterized the composites in detail. The prepared composites were applied as adsorbents for the removal of OTC, TC and CTC from aqueous solutions. The contents are important for the removal of TCs from aqueous solutions. The manuscript is well organized. After reading the manuscript, I think it can be accepted for publication after revision.

Special comments:

1. It is necessary to give the experimental conditions for the sorption capacities as the sorption capacity is dependent on solution conditions.
2. In the Introduction section, several critical reviews should be added in the revised form such as *The Innovation*, 2021, 2(1), 100076; *Science of Total Environment*. 2021, 760, 143333; *Chemosphere*. Doi.org/10.1016/j.chemosphere.2020.128576.
3. How about the stability of the composites, especially at low pH values?
4. Please check references 64 and 65 to update the volume and page number.
5. TOC is too complicated. I suggest the authors to add one simple and clear TOC in the revised form.
6. The authors mentioned four mechanisms, I suggest the authors to discuss the mechanism more detailly in the revised form such as *Journal of Hazardous Materials*. 2020, 393, 122353; *Chemosphere*, 2020, 246, 125753.
7. The reason for the sorption amounts of OTC>TC>CTC should be discussed more detailly in the revised form.

Review form: Reviewer 2

Is the manuscript scientifically sound in its present form?

Yes

Are the interpretations and conclusions justified by the results?

Yes

Is the language acceptable?

Yes

Do you have any ethical concerns with this paper?

No

Have you any concerns about statistical analyses in this paper?

Yes

Recommendation?

Major revision is needed (please make suggestions in comments)

Comments to the Author(s)

Effective removal of tetracycline antibiotics from wastewater using practically applicable iron(III)-loaded cellulose nanofibers

Manuscript ID RSOS-210336

The removal of TCs is of environmental importance due to their toxicity. The work in this manuscript shows enough novelty and scientific soundness to meet the requirement for

publishing in Royal Society Open Science after major revision. There are some problems in this work as follow:

1. The highlights do not contain sufficient novel features of the work and should be rewritten.
2. What is the novelty of the work? Mention the novelty of the work clearly
3. What scientific problems can be solved by the design? The statements about novelty of the manuscript should be stressed if any.
4. Sorbents for TCs should be updated. Recent sorbents reported should be described. Authors should consult and include the following papers (such as Applied Surface Science 543 (2021) 148810; Journal of Alloys and Compounds 863, (2021) 158475; Chemosphere 255 (2020) 126917; Environmental Research 191 (2020) 110089; Science of the Total Environment, 651 (2019) 1653-1660) in the revised manuscript.
5. The para. from line 113 to 123 can be described in section 2.6 and is independent of the novelty and scientific problems of this work.
6. The detail conditions and process of TCs determination by spectrophotometric method should be given. The concentrations of TCs were determined usually by HPLC.
7. Characterizations of the materials are too simple to present its physicochemical properties. More information from TG, TEM and XRD analysis, so on could be offered if possible.
8. From the images of SEM, no nanofibers were found. Why? The SEM images of iron(III)-loaded cellulose nanofibers should be exhibited in the text.
9. The linear equations of the kinetic, thermodynamics and isothermal models and should be described in text.
10. The E values from Dubinin-Radushkevich isotherm should be calculated. Please read the following papers: Environmental Research, 195 (2021) 110752.
11. Table S3 should be exhibited in the text.
12. The English writing should be improved as a research manuscript.
13. The origin references of isotherms and kinetic models should be provided. There are some mistakes in References (such as No.62).

Decision letter (RSOS-210336.R0)

Dear Miss Lu:

Title: Effective removal of tetracycline antibiotics from wastewater using practically applicable iron(III)-loaded cellulose nanofibers

Manuscript ID: RSOS-210336

The editor assigned to your manuscript has now received comments from reviewers. We would like you to revise your paper in accordance with the referee and Subject Editor suggestions which can be found below (not including confidential reports to the Editor). Please note this decision does not guarantee eventual acceptance.

Please submit your revised paper before 02-May-2021. Please note that the revision deadline will expire at 00.00am on this date. If we do not hear from you within this time then it will be assumed that the paper has been withdrawn. In exceptional circumstances, extensions may be

possible if agreed with the Editorial Office in advance. We do not allow multiple rounds of revision so we urge you to make every effort to fully address all of the comments at this stage. If deemed necessary by the Editors, your manuscript will be sent back to one or more of the original reviewers for assessment. If the original reviewers are not available we may invite new reviewers.

On behalf of the Subject Editor Professor Anthony Stace and the Associate Editor Dr Nadia Martinez Villegas.

RSC Associate Editor:

Comments to the Author:

The research presented in this draft is original and of interest to RSOS audience, however more details on the experimental conditions as well as a deeper material characterization and explanation on sorption mechanisms and trends are needed. Please read carefully each of the comments from the reviewers and address each of them.

RSC Subject Editor:

Comments to the Author:

(There are no comments.)

Reviewers' Comments to Author:

Reviewer: 1

Comments to the Author(s)

In this manuscript, the authors prepared Fe(III)@CNFs and characterized the composites in detail. The prepared composites were applied as adsorbents for the removal of OTC, TC and CTC from aqueous solutions. The contents are important for the removal of TCs from aqueous solutions. The manuscript is well organized. After reading the manuscript, I think it can be accepted for publication after revision.

Special comments:

1. It is necessary to give the experimental conditions for the sorption capacities as the sorption capacity is dependent on solution conditions.
2. In the Introduction section, several critical reviews should be added in the revised form such as *The Innovation*, 2021, 2(1), 100076; *Science of Total Environment*. 2021, 760, 143333; *Chemosphere*. Doi.org/10.1016/j.chemosphere.2020.128576.
3. How about the stability of the composites, especially at low pH values?
4. Please check references 64 and 65 to update the volume and page number.
5. TOC is too complicated. I suggest the authors to add one simple and clear TOC in the revised form.
6. The authors mentioned four mechanisms, I suggest the authors to discuss the mechanism more detailly in the revised form such as *Journal of Hazardous Materials*. 2020, 393, 122353; *Chemosphere*, 2020, 246, 125753.
7. The reason for the sorption amounts of OTC>TC>CTC should be discussed more detailly in the revised form.

Reviewer: 2

Comments to the Author(s)

Effective removal of tetracycline antibiotics from wastewater using practically applicable iron(III)-loaded cellulose nanofibers

Manuscript ID RSOS-210336

The removal of TCs is of environmental importance due to their toxicity. The work in this manuscript shows enough novelty and scientific soundness to meet the requirement for publishing in Royal Society Open Science after major revision. There are some problems in this work as follow:

1. The highlights do not contain sufficient novel features of the work and should be rewritten.
2. What is the novelty of the work? Mention the novelty of the work clearly
3. What scientific problems can be solved by the design? The statements about novelty of the manuscript should be stressed if any.
4. Sorbents for TCs should be updated. Recent sorbents reported should be described. Authors should consult and include the following papers (such as *Applied Surface Science* 543 (2021) 148810; *Journal of Alloys and Compounds* 863, (2021) 158475; *Chemosphere* 255 (2020) 126917; *Environmental Research* 191 (2020) 110089; *Science of the Total Environment*, 651 (2019) 1653-1660) in the revised manuscript.
5. The para. from line 113 to 123 can be described in section 2.6 and is independent of the novelty and scientific problems of this work.
6. The detail conditions and process of TCs determination by spectrophotometric method should be given. The concentrations of TCs were determined usually by HPLC.
7. Characterizations of the materials are too simple to present its physicochemical properties. More information from TG, TEM and XRD analysis, so on could be offered if possible.
8. From the images of SEM, no nanofibers were found. Why? The SEM images of iron(III)-loaded cellulose nanofibers should be exhibited in the text.
9. The linear equations of the kinetic, thermodynamics and isothermal models and should be described in text.
10. The E values from Dubinin-Radushkevich isotherm should be calculated. Please read the following papers: *Environmental Research*, 195 (2021) 110752.
11. Table S3 should be exhibited in the text.

12. The English writing should be improved as a research manuscript.

13. The origin references of isotherms and kinetic models should be provided. There are some mistakes in References (such as No.62).

Author's Response to Decision Letter for (RSOS-210336.R0)

See Appendix A.

RSOS-210336.R1 (Revision)

Review form: Reviewer 1

Is the manuscript scientifically sound in its present form?

Yes

Are the interpretations and conclusions justified by the results?

Yes

Is the language acceptable?

Yes

Do you have any ethical concerns with this paper?

No

Have you any concerns about statistical analyses in this paper?

No

Recommendation?

Accept as is

Comments to the Author(s)

The authors revised the manuscript carefully and I recommend for publication.

Review form: Reviewer 2

Is the manuscript scientifically sound in its present form?

Yes

Are the interpretations and conclusions justified by the results?

Yes

Is the language acceptable?

Yes

Do you have any ethical concerns with this paper?

No

Have you any concerns about statistical analyses in this paper?

No

Recommendation?

Accept as is

Comments to the Author(s)

I consider that the presented work can be published .

Decision letter (RSOS-210336.R1)

Dear Miss Lu:

Title: Effective removal of tetracycline antibiotics from wastewater using practically applicable iron(III)-loaded cellulose nanofibers

Manuscript ID: RSOS-210336.R1

It is a pleasure to accept your manuscript in its current form for publication in Royal Society Open Science. The chemistry content of Royal Society Open Science is published in collaboration with the Royal Society of Chemistry.

On behalf of the Subject Editor Professor Anthony Stace and the Associate Editor Dr Nadia Martinez Villegas.

RSC Associate Editor:

Comments to the Author:

The authors have satisfactorily addressed every single comment and remark from the reviewers.
The manuscript can now be accepted.

RSC Subject Editor:

Comments to the Author:

(There are no comments.)

Reviewer(s)' Comments to Author:

Reviewer: 1

Comments to the Author(s)

The authors revised the manuscript carefully and I recommend for publication.

Reviewer: 2

Comments to the Author(s)

I consider that the presented work can be published .

Appendix A

Response to Referee: 1

Comments to the Author(s)

In this manuscript, the authors prepared Fe(III)@CNFs and characterized the composites in detail. The prepared composites were applied as adsorbents for the removal of OTC, TC and CTC from aqueous solutions. The contents are important for the removal of TCs from aqueous solutions. The manuscript is well organized. After reading the manuscript, I think it can be accepted for publication after revision.

Response: We greatly appreciate your constructive comments and we have prepared a detailed reply to your comments.

Special comments:

1. It is necessary to give the experimental conditions for the sorption capacities as the sorption capacity is dependent on solution conditions.

Response: Thanks a lot for your comment. In order to clarify the experimental conditions more clearly, corresponding experimental conditions below each figure have been added (Figures and tables caption and lines 60-61 in the revised supplementary material).

2. In the Introduction section, several critical reviews should be added in the revised form such as The Innovation, 2021, 2(1), 100076; Science of Total Environment. 2021, 760, 143333; Chemosphere. Doi.org/10.1016/j.chemosphere.2020.128576.

Response: Thanks a lot for your comment. We have carefully read these critical reviews and added them in the **Introduction** section (Lines 66 and 79 in the revised manuscript).

The modified BiOCl is a promising photocatalyst in photocatalysis and tetracycline

antibiotics (TCs) can be photocatalyzed by the modified BiOCl [1]. For example, Al-doped BiOCl [2], Y³⁺-doped BiOCl [3], and BiOCl/TiO₂ [4] can reach photocatalytic efficiency of over 84% in a short time. The application of carbon materials derived from metal organic frameworks (MOFs) have become a rapidly expanding research field [5]. For example, Co₃O₄/CNTs [6] and MIL/CNT-Fe [7] performed well as catalysts in the applications of removing oxytetracycline (OTC) and tetracycline (TC) respectively. Covalent organic frameworks-based (COF) materials as superior adsorbents for the efficient removal of organic pollutants [8]. A novel COF-based material named as NCCT reached the high removal capacities of TC (388.52 mg/g) [9].

References

- [1] Yao L, Yang H, Chen Z, Qiu M, Hu B, Wang X. 2020 Bismuth oxychloride-based materials for the removal of organic pollutants in wastewater. *Chemosphere*, 128576. (doi:10.1016/j.chemosphere.2020.128576)
- [2] Zhang J et al. 2020 Enhanced photocatalytic degradation of tetracycline hydrochloride by Al-doped BiOCl microspheres under simulated sunlight irradiation. *Chem. Phys. Lett.* 750, 137483. (doi:10.1016/j.cplett.2020.137483)
- [3] Zhong S, Wang X, Wang Y, Zhou F, Li J, Liang S, Li C. 2020 Preparation of Y³⁺-doped BiOCl photocatalyst and its enhancing effect on degradation of tetracycline hydrochloride wastewater. *J. Alloys Compd.* 843, 155598. (doi:10.1016/j.jallcom.2020.155598)
- [4] Bao S, Liang H, Li C, Bai J. 2020 A heterostructure BiOCl nanosheets/TiO₂ hollow-tubes composite for visible light-driven efficient photodegradation antibiotic. *J. Photochem. Photobiol. A Chem.* 397, 112590. (doi:10.1016/j.jphotochem.2020.112590)

- [5] Hao M, Qiu M, Yang H, Hu B, Wang X. 2021 Recent advances on preparation and environmental applications of MOF-derived carbons in catalysis. *Sci. Total Environ.* 760, 143333. (doi:10.1016/j.scitotenv.2020.143333)
- [6] Liu D, Li M, Li X, Ren F, Sun P, Zhou L. 2020 Core-shell Zn/Co MOFs derived Co₃O₄/CNTs as an efficient magnetic heterogeneous catalyst for persulfate activation and oxytetracycline degradation. *Chem. Eng. J.* 387, 124008. (doi:10.1016/j.cej.2019.124008)
- [7] Yang C et al. 2019 Augmenting Intrinsic Fenton-Like Activities of MOF-Derived Catalysts via N-Molecule-Assisted Self-catalyzed Carbonization. *Nano-Micro Lett.* 11, 1–13. (doi:10.1007/s40820-019-0319-4)
- [8] Liu X et al. 2021 Orderly Porous Covalent Organic Frameworks-based Materials: Superior Adsorbents for Pollutants Removal from Aqueous Solutions. *Innov.* 2, 100076. (doi:10.1016/j.xinn.2021.100076)
- [9] Li Z, Liu Y, Zou S, Lu C, Bai H, Mu H, Duan J. 2020 Removal and adsorption mechanism of tetracycline and cefotaxime contaminants in water by NiFe₂O₄-COF-chitosan-terephthalaldehyde nanocomposites film. *Chem. Eng. J.* 382, 123008. (doi:10.1016/j.cej.2019.123008)

3. How about the stability of the composites, especially at low pH values?

Response: We greatly appreciate the reviewer's comments. An experiment has been conducted to test stability of the composites at acidic, neutral, and alkaline pH respectively. The experimental conditions and results were added (Lines 166-170 and 345-350 in the revised manuscript).

50 mg Fe(III)@CNFs were added into 100 mL aqueous solution with pH 3.07, 7.05 and

12.00 respectively and reacted for 24h. The iron content in the leachate was tested and analyzed by atomic absorption spectrophotometer (GGX-600, HAIGUANG INSTRUMENT). The specific parameters were designed as follows: air flow rate is 7.6 L/min, acetylene flow rate is 1.3 L/min, 370 V, 10 mA, and measurement was performed at a wavelength of 248.3nm. The experimental results showed that the iron content of Fe(III)@CNFs at pH 7.05 and 12.00 was less than 0.03 mg/L (detection limit), while the iron content at pH 3.07 was 0.241 mg/L. Although Fe(III)@CNFs has iron leaching at low pH, the leaching amount is very low. In addition, the adsorbent is mostly used in sewage with neutral pH, so the stability of the composites can be guaranteed.

4. Please check references 64 and 65 to update the volume and page number.

Response: We greatly appreciate the reviewer's comments. We are very sorry that the volume and page number were not updated when inserting references 64 and 65 in the manuscript. We have updated the volume and page number as follows (Lines 800-805 in the revised manuscript):

Liu P, Wang Q, Zheng C, He C. 2017 Sorption of Sulfadiazine, Norfloxacin, Metronidazole, and Tetracycline by Granular Activated Carbon: Kinetics, Mechanisms, and Isotherms. *Water. Air. Soil Pollut.* 228, 129. (doi:10.1007/s11270-017-3320-x)

Zhang B, Zhang H, Li X, Lei X, Li C, Yin D, Fan X, Zhang Q. 2013 Synthesis of BSA/Fe₃O₄ magnetic composite microspheres for adsorption of antibiotics. *Mater. Sci. Eng. C* 33, 4401-4408. (doi:10.1016/j.msec.2013.06.038)

5. TOC is too complicated. I suggest the authors to add one simple and clear TOC in the revised form.

Response: Thanks a lot for your constructive comments. We have simplified the TOC and made it clearer (Revised-graphical abstract). The revised TOC also can be found in the figure R1.

Figure R1. Revised TOC.

6. The authors mentioned four mechanisms, I suggest the authors to discuss the mechanism more detailly in the revised form such as *Journal of Hazardous Materials*. 2020, 393, 122353; *Chemosphere*, 2020, 246, 125753.

Response: We greatly appreciate your comments. We have carefully read articles you recommended and the mechanism was discussed more detailly (Lines 529-533 and 535-538 in the manuscript).

TCs' electron-rich ketone, carboxyl, amino, and hydroxyl groups contribute to their strong tendency to complex with metals. In Chen et al.'s study [10], TC, chlorotetracycline (CTC) and OTC adsorb strongly to aluminum oxide (Al_2O_3). Adsorption sites were also simulated force fields, which protonated carbonyl and amino groups of TCs that can bind cations effectively and form complexes [11]. The Fe(III) ions with TC, CTC and OTC form the stable complexes with the phenolic β -diketone structure of TCs [12]. In our study, the

presence and absence of Fe(III) has a significant effect on the adsorption of TCs, thus the surface complexation between Fe(III) and TCs can be verified. Fe(III)@CNFs contain carboxyl groups and hydroxyl functional groups that can lead to the formation of a hydrogen bond with phenolic hydroxyl, amine, hydroxyl, and ketene moieties of TCs [13]. In addition, the FTIR spectra of Fe(III)@CNFs before and after TCs adsorption also proved the formation of hydrogen bond. In addition, the adsorption tendency of the Fe(III)@CNFs to TCs indicates the electrostatic interaction at different pH. Strong changes in adsorption capacities at acidic and alkaline pH proved the electrostatic interaction occupies a significant position in the adsorption process. Therefore, the adsorption amounts of TCs by Fe(III)@CNFs composites were promoted significantly compared with CNFs due to the synergistic interactions of the surface complexation, the hydrogen bond, the electrostatic interactions and the van der Waals forces.

References

- [10] Chen WR, Huang CH. 2010 Adsorption and transformation of tetracycline antibiotics with aluminum oxide. *Chemosphere* 79, 779–785. (doi:10.1016/j.chemosphere.2010.03.020)
- [11] Zhong X, Lu Z, Liang W, Hu B. 2020 The magnetic covalent organic framework as a platform for high-performance extraction of Cr(VI) and bisphenol a from aqueous solution. *J. Hazard. Mater.* 393, 122353. (doi:10.1016/j.jhazmat.2020.122353)
- [12] Song YX, Chen S, You N, Fan HT, Sun LN. 2020 Nanocomposites of zero-valent Iron@Activated carbon derived from corn stalk for adsorptive removal of tetracycline antibiotics. *Chemosphere* 255, 126917. (doi:10.1016/j.chemosphere.2020.126917)
- [13] Wei X, Zhang R, Zhang W, Yuan Y, Lai B. 2019 High-efficiency adsorption of

tetracycline by the prepared waste collagen fiber-derived porous biochar. RSC Adv. 9, 39355–39366. (doi:10.1039/c9ra07289f)

7. The reason for the sorption amounts of OTC>TC>CTC should be discussed more detailly in the revised form.

Response: We greatly appreciate your constructive comments. The reason for the sorption amounts of OTC>TC>CTC has been discussed more detailly (Lines 491-500 in the revised manuscript).

The difference of chemical structure among OTC, TC and CTC is the substituent. The adding amount of hydroxyl in OTC resulted in the increase of combination sites with Fe(III)@CNFs. Fe(III)@CNFs contain carboxyl groups and hydroxyl functional groups that can lead to the formation of a hydrogen bond with hydroxyl of OTC [13]. Thus, the adsorption capacities of Fe(III)@CNFs to OTC is higher than adsorption capacity to TC. In addition, the additional chlorine atom at C7 position in CTC have the potential effect on electric density of the whole molecules and enhanced polarity of CTC's functional groups [10], therefore the adsorption capacity of Fe(III)@CNFs to CTC is just 232.56 mg/g. Comparing to CTC, Fe(III)@CNFs can form the more complex with OTC and TC.

Response to Referee: 2

Comments to the Author(s)

The removal of TCs is of environmental importance due to their toxicity. The work in this manuscript shows enough novelty and scientific soundness to meet the requirement for publishing in Royal Society Open Science after major revision. There are some problems in

this work as follow:

Response: Thanks a lot for your constructive comments and we have prepared a detailed reply to your comments.

1. The highlights do not contain sufficient novel features of the work and should be rewritten.

Response: Thanks a lot for your comments. The novel features of the work have been carefully summarized and the highlights have been rewritten as follows (Revised-highlights):

(1) The renewable and non-sintered Fe(III)@CNFs derived from bamboo were optimized.

(2) TC, CTC, and OTC can be adsorbed by Fe(III)@CNFs simultaneously and effectively.

(3) Fe(III)@CNFs exhibited stable adsorption and excellent tolerance in a wide pH range.

(4) Fe(III)@CNFs with advantages of excellent reproducibility and firm Fe(III) anchoring.

(5) Adsorption of TCs fitted well with Langmuir and pseudo-second-order models.

2. What is the novelty of the work? Mention the novelty of the work clearly.

Response: Thanks a lot for your comments. The novelty of the work have been mentioned (Lines 84-86, 103-104, 573, 580-582, and 586-588 in the revised manuscript).

Bamboo grows fast and the bamboo wastes are abundant in Sichuan province of China. The development and utilization of bamboo resources is conducive to broadening the application of bamboo resource products. The non-toxic and completely biodegradable cellulose within bamboo is one of the most abundant agricultural polysaccharide wastes in

the world, and can be processed into CNFs by mechanical shearing method in this study. Fe(III) has the advantages of being environmentally friendly, almost harmless to organisms and being able to complex with TCs. Therefore, the renewable and non-sintered Fe(III)@CNFs were developed and optimized to adsorb TC, CTC, and OTC from different metal and Fe(III) salts. The manufactured Fe(III)@CNFs exhibited stable adsorption and excellent tolerance in a wide pH range that tend to reduce the cost of pH adjustment. In addition, The strong reproducibility, firm Fe(III) fixation and the characteristics of not being easily affected by coexisting ions greatly broaden the application possibilities of Fe(III)@CNFs in actual sewage treatment.

3. What scientific problems can be solved by the design? The statements about novelty of the manuscript should be stressed if any.

Response: We greatly appreciate your comments. TCs are ubiquitously detected in wastewater and exist in the aquatic environment persistently because they are incompletely metabolized in organisms, are difficult to degrade, and have high hydrophilicity and low volatility. TCs' electron-rich ketone, carboxyl, amino, and hydroxyl groups contribute to their strong tendency to complex with metals. CNFs can be used as a framework to loading Fe(III) to remove TC, CTC, and OTC through the synergistic interactions of the surface complexation, the hydrogen bond, the electrostatic interactions and the van der Waals forces. Through the design, a renewable and non-sintered adsorbent with stable adsorption performance is desired to solve the scientific problem of simultaneous adsorption of TC, CTC and OTC. The statements about novelty of the manuscript have been stressed (Lines 122-124 in the revised manuscript).

4. Sorbents for TCs should be updated. Recent sorbents reported should be described. Authors should consult and include the following papers (such as *Applied Surface Science* 543 (2021) 148810; *Journal of Alloys and Compounds* 863, (2021) 158475; *Chemosphere* 255 (2020) 126917; *Environmental Research* 191 (2020) 110089; *Science of the Total Environment*, 651 (2019) 1653-1660) in the revised manuscript.

Response: We greatly appreciate your comments. We have read these papers and sorbents for TCs have been updated (Lines 79-82 in the revised manuscript).

The nanocomposites of reduced graphene oxide with the ZrO_2 ($ZrO_2@rGO$) [14], the hybrid nanocomposites of zero-valent iron loaded the activated carbon ($ZVI@ACCS$) [12], and the hybrid sorbent of α -iron oxide/reduced graphene oxide ($\alpha-Fe_2O_3@RGO$) [15] have been used to adsorb TCs. In addition, the graphene nanoplatelet-based diffusion gradients in thin films (G-DGT) [16] and the nanosized ZnO-based diffusion gradients in thin films (nanoZnO-DGT) [17] were developed for in situ sampling of TC, CTC and OTC in aquatic environment. $ZrO_2@rGO$ has been applied to OTC adsorption and demonstrated a remarkably selective uptake of OTC among TCs due to their higher affinity towards OTC. $ZVI@ACCS$ derived from the corn stalk and $\alpha-Fe_2O_3@RGO$ have been used to remove TCs from aqueous solution and the adsorption amount of CTC is higher than that of TC and OTC. G-DGT is more suitable than the nanoZnO-DGT as an effective DGT binding agent to provide good representative in situ sampling of TCs.

References

[14] Hao D, Song YX, Zhang Y, Fan HT. 2021 Nanocomposites of reduced graphene oxide with pure monoclinic- ZrO_2 and pure tetragonal- ZrO_2 for selective adsorptive removal of

oxytetracycline. Appl. Surf. Sci. 543, 148810. (doi:10.1016/j.apsusc.2020.148810)

[15] Zou SJ, Chen YF, Zhang Y, Wang XF, You N, Fan HT. 2021 A hybrid sorbent of α -iron oxide/reduced graphene oxide: Studies for adsorptive removal of tetracycline antibiotics. J. Alloys Compd. 863, 158475. (doi:10.1016/j.jallcom.2020.158475)

[16] You N, Chen S, Wang Y, Fan HT, Sun LN, Sun T. 2020 In situ sampling of tetracycline antibiotics in culture wastewater using diffusive gradients in thin films equipped with graphene nanoplatelets. Environ. Res. 191, 110089. (doi:10.1016/j.envres.2020.110089)

[17] You N, Yao H, Wang Y, Fan HT, Wang CS, Sun T. 2019 Development and evaluation of diffusive gradients in thin films based on nano-sized zinc oxide particles for the in situ sampling of tetracyclines in pig breeding wastewater. Sci. Total Environ. 651, 1653–1660. (doi:10.1016/j.scitotenv.2018.09.323)

5. The para. from line 113 to 123 can be described in section 2.6 and is independent of the novelty and scientific problems of this work.

Response: We greatly appreciate your comments. We have reorganized the structure of **Section 2.6** and introduced the regeneration method in **Section 3.4** (Lines 394-396 in the revised manuscript).

6. The detail conditions and process of TCs determination by spectrophotometric method should be given. The concentrations of TCs were determined usually by HPLC.

Response: Thanks a lot for your comments. In this study, the concentration of TCs is at the mg/L level and spectrophotometric method can be used. The ultraviolet spectrum of TC, CTC and OTC was scanned in the wavelength range of 190-400 nm using ultraviolet-visible spectrophotometer (N4, INESA, China) and the wavelength corresponding to the

strongest absorbance value was obtained (Figure R2). In addition, the concentration of TCs has an excellent linear relationship with absorbance at the corresponding wavelength (Figure R2). Therefore, the concentration of TCs was tested by spectrophotometric method. The detailed conditions and process of TCs determination by spectrophotometric method have been given (Lines 186-190).

Figure R2. The ultraviolet spectrum (a) and standard curve (b) of TCs.

7. Characterizations of the materials are too simple to present its physicochemical properties. More information from TG, TEM and XRD analysis, so on could be offered if possible.

Response: We greatly appreciate your constructive comments. We have added thermogravimetric analysis-differential scanning calorimetry (TGA-DSC) and X-ray diffraction (XRD) analysis to present the physicochemical properties of materials further. Since sample preparation of TEM requires the sample to be dispersed in a solvent, and in theory, the thinner the sample is, the easier it is to observe, it is necessary to grind the sample sufficiently and disperse it in the solvent. We have tried to test our materials for TEM. However, it is difficult to meet the TEM sample preparation requirements of the thickness of 100-200 nm and the time for returning the manuscript is limited, thus TEM has not been

completed. We will pay attention to this test in future research.

TGA-DSC (TGA/DSC3+/1600, METTLER TOLEDO) was performed under 0-1000°C to exam the thermal stability of CNFs and Fe(III)@CNFs (Figure R3). The specific parameters are designed as follows: nitrogen atmosphere, heating rate 20 °C/min. TGA-DSC was detected by Analytical and Testing Center, Sichuan University, China. The TGA of the CNFs and Fe(III)@CNFs heated in the air, the weight loss below 250 °C is attributed to the evaporation of adsorbed water [15]. A sharp weight loss (87.67%) of CNFs from 250° to 400°C and a gradual weight loss of Fe(III)@CNFs ranged 250–850 °C were observed because of the decomposition of materials. At different temperatures, the weight loss of Fe(III)@CNFs was always less than that of CNFs, and the final weight of Fe(III)@CNFs can be maintained at about 50%, while the final weight of CNFs was as low as about 10%. The results of TGA-DSC indicated that the loading of Fe(III) improved the thermal stability of the CNFs. The information of TGA-DSC was offered (Lines 166-170 and 319-327 in the revised manuscript).

Figure R3. TGA-DSC of the CNFs (a) and Fe(III)@CNFs (b).

The crystalline phases in CNFs and Fe(III)@CNFs samples were tested and analyzed by XRD (EMPYREAN, PANalytical) (Figure R4). The specific parameters are designed as

follows: Cu target, 40 kV, 40 mA, within the 2θ angle range of $3\sim 90^\circ$, XRD patterns were recorded at scan step size of 0.039 and time per step of 20.91. XRD was detected by Analytical and Testing Center, Sichuan University, China. CNFs contain three peaks located at 15.6° , 22.6° and 34.5° , which belong to the typical spectrum of cellulose [18]. However, the peaks disappeared after Fe(III) loading, indicating the crystalline form of cellulose was destroyed after modification. The information of XRD was offered (Lines 173-178 and 312-316).

Figure R4. XRD patterns of the CNFs and Fe(III)@CNFs.

References

[18] Sun J, Cui L, Gao Y, He Y, Liu H, Huang Z. 2021 Environmental application of magnetic cellulose derived from *Pennisetum sinense* Roxb for efficient tetracycline removal. *Carbohydr. Polym.* 251, 117004. (doi:10.1016/j.carbpol.2020.117004)

8. *From the images of SEM, no nanofibers were found. Why? The SEM images of iron(III)-loaded cellulose nanofibers should be exhibited in the text.*

Response: Thanks a lot for your comments. Nanofibers refer to the average pore size

of the material at the nanometer level, and the fibrous material is not noticeable due to the insufficient magnification of the SEM image. In Figure S1 of Zhang's article [19], it can be seen that the material is obviously fibrous at magnification of 80,000×. The SEM images of iron(III)-loaded cellulose nanofibers have been exhibited in the text (Figure 2 in the revised manuscript).

References

[19] Zhang X, Zhao J, He X, Li Q, Ao C, Xia T, Zhang W, Lu C, Deng Y. 2018 Mechanically robust and highly compressible electrochemical supercapacitors from nitrogen-doped carbon aerogels. *Carbon N. Y.* 127, 236–244. (doi:10.1016/j.carbon.2017.10.083)

9. *The linear equations of the kinetic, thermodynamics and isothermal models and should be described in text.*

Response: Thanks a lot for your comments. The linear equations of the kinetic, thermodynamics and isothermal models have been described (Lines 7-52 in the revised supplementary material). In addition, this is explained in the revised manuscript (Lines 212-214).

10. *The E values from Dubinin-Radushkevich isotherm should be calculated. Please read the following papers: Environmental Research, 195 (2021) 110752.*

Response: We greatly appreciate your comments. We have read the paper and the E values from Dubinin-Radushkevich isotherm have been calculated (Table 2 in the revised manuscript) according the equation (R1) as follows [20]:

$$E = -(2K_{DR})^{-1/2} \quad (R1)$$

References

[20] Zou SJ, Ding BH, Chen YF, Fan HT. 2021 Nanocomposites of graphene and zirconia for adsorption of organic-arsenic drugs: Performances comparison and analysis of adsorption behavior. *Environ. Res.* 195, 110752. (doi:10.1016/j.envres.2021.110752)

11. Table S3 should be exhibited in the text.

Response: Thanks a lot for your comments. The thermodynamic parameters for the adsorption of tetracycline antibiotics(TCs) on Fe(III)@CNFs have been exhibited in the revised manuscript (Table 4).

12. The English writing should be improved as a research manuscript.

Response: We greatly appreciate your comments. The revised manuscript have been carefully edited by a native English-speaking editor of Editeg, and the grammar, spelling, and punctuation have been verified and corrected where needed. We have read the manuscript several times and made some revisions in the revised manuscript.

13. The origin references of isotherms and kinetic models should be provided. There are some mistakes in References (such as No.62).

Response: We greatly appreciate your comments. The origin references of kinetic models (the Pseudo first order model [21], the Pseudo second order model [22], Morris-Weber intra-particle diffusion [23] and Simple Elovich model [24]) and isotherms models (the Langmuir model [25], the Freundlich model [26], the Temkin model [27] and the Dubinin-Radushkevich (D-R) model [28]) have been provided (Lines 208-211 in the revised manuscript). In addition, the errors in reference 62 have been corrected (Lines 736-737).

References

[21] Lagergren S. 1898 About the theory of so-called adsorption of soluble substances. K. -

Sven. Vetenskapsakademiens Handl. 24, 1–39.

[22] Ho YS, McKay G. 1999 Pseudo-second-order model for sorption processes. *Process Biochem.* 34, 451–465. (doi: 10.1016/S0032-9592(98)00112-5)

[23] Weber WJ, Morris JC. 1963 Kinetics of adsorption of carbon from solutions. *J. Sanit. Eng. Div. Am. Soc. Civ. Eng.* 89, 31–63.

[24] Sparks DL. 1989 *Kinetics of Soil Chemical Processes*, Academic Press, Inc., New York.

[25] Langmuir I. 1918 The adsorption of gases on plane surfaces of glass, mica and platinum. *J. Am. Chem. Soc.* 40, 1361–1403. (doi: 10.1021/ja02242a004)

[26] Freundlich HMF. 1906 Über die absorption in lösungen. *Z Phys Chem* 57, 385–470.

[27] Temkin MI. 1940 Kinetics of ammonia synthesis on promoted iron catalysts. *Acta Physiochim. URSS* 12, 327–356.

[28] Dubinin MM, Radushkevich LV. 1947 Equation of the characteristic curve of activated charcoal. *Chem Zentr* 1, 875.